# In situ ozone production is highly sensitive to volatile organic compounds in Delhi, India

Beth S. Nelson[1], Gareth J. Stewart[1], Will S. Drysdale[1,2], Mike J. Newland[1], Adam R. Vaughan[1], Rachel E. Dunmore[1], Pete M. Edwards[1], Alastair C. Lewis[1,2], Jacqueline F. Hamilton[1], W. Joe Acton[3a], C. Nicholas Hewitt[3], Leigh R. Crilley[4b], Mohammed S. Alam[4], Ülkü A. Şahin[5], David C. S. Beddows[2,4], William J. Bloss[4], Eloise Slater[6], Lisa K. Whalley[6,7], Dwayne E. Heard[6], James M. Cash[8], Ben Langford[8], Eiko Nemitz[8], Roberto Sommariva[4], Sam Cox[9], Shivani[10c], Ranu Gadi[10], Bhola R. Gurjar[11], James R. Hopkins[1,2], Andrew R. Rickard[1,2], James D. Lee[1,2]

[1]Wolfson Atmospheric Chemistry Laboratories, University of York, Heslington, York, UK
[2]National Centre for Atmospheric Science, University of York, Heslington, York, UK
[3]Lancaster Environment Centre, Lancaster University, Lancaster, UK
[4]School of Geography, Earth and Environmental Sciences, University of Birmingham, Birmingham, UK
[5]Istanbul University-Cerrahpasa, Engineering Faculty, Environmental Engineering Department, Avcilar, Istanbul, Turkey
[6]School of Chemistry, University of Leeds, Leeds, UK
[7]National Centre for Atmospheric Science, University of Leeds, Leeds, UK
[8]UK Centre for Ecology and Hydrology, Edinburgh, UK
[9]Research Software Engineering Team, University of Leicester, Leicester, UK
[10]Indira Gandhi Delhi Technical University for Women, Delhi, India
[11]Indian Institute of Technology, Roorkee, India
[a]now at: School of Geography, Earth and Environmental Sciences, University of Birmingham, Birmingham, UK
[b]now at: Department of Chemistry, York University, Toronto, Ontario, Canada
[c]now at: Department of Chemistry, Miranda House, Delhi University, Delhi, India

*Correspondence to*: Beth S. Nelson (bsn502@york.ac.uk)

**Abstract.** The Indian megacity of Delhi suffers from some of the poorest air quality in the world. While ambient $NO_2$ and particulate matter (PM) concentrations have received considerable attention in the city, high ground level ozone ($O_3$) concentrations are an often overlooked component of pollution. $O_3$ can lead to significant ecosystem damage, agricultural crop losses, and adversely affect human health. During October 2018, concentrations of speciated non-methane hydrocarbons volatile organic compounds ($C_2 – C_{13}$), oxygenated volatile organic compounds (o-VOCs), NO, $NO_2$, HONO, CO, $SO_2$, $O_3$, and photolysis rates, were continuously measured at an urban site in Old Delhi. These observations were used to constrain a detailed chemical box model utilising the Master Chemical Mechanism v3.3.1. VOCs and $NO_x$ (NO + $NO_2$) were varied in the model to test their impact on local $O_3$ production rates, $P(O_3)$, which revealed a VOC-limited chemical regime. When only $NO_x$ concentrations were reduced, a significant increase in $P(O_3)$ was observed, thus VOC co-reduction approaches must also be considered in pollution abatement strategies. Of the VOCs examined in this work, mean morning $P(O_3)$ rates were most sensitive to monoaromatic compounds, followed by monoterpenes and alkenes, where halving their concentrations in the model led to a 15.6 %, 13.1 % and 12.9 % reduction in $P(O_3)$, respectively. $P(O_3)$ was not sensitive to direct changes in aerosol surface area but was very sensitive to changes in photolysis rates, which may be influenced by future changes in PM

concentrations. VOC and $NO_x$ concentrations were divided into emission source sectors, as described by the EDGAR v5.0 Global Air Pollutant Emissions and EDGAR v4.3.2_VOC_spec inventories, allowing for the impact of individual emission sources on $P(O_3)$ to be investigated. Reducing road transport emissions only, a common strategy in air pollution abatement

strategies worldwide, was found to increase $P(O_3)$, even when the source was removed in its entirety. Effective reduction in $P(O_3)$ was achieved by reducing road transport along with emissions from combustion for manufacturing and process emissions. Modelled $P(O_3)$ reduced by $\sim 20$ ppb $h^{-1}$ when these combined sources were halved. This study highlights the importance of reducing VOCs in parallel with $NO_x$ and PM in future pollution abatement strategies in Delhi.

## 1 Introduction

The majority of the world's population now live in urban areas. This is projected to increase from 55 % in 2018 to 68 % of the global population by 2030, with 90 % of this growth occurring in Asia and Africa (Molina, 2021; United Nations, 2019; United Nations, 2018). Rapidly increasing industrialisation and urbanisation, coinciding with fast population growth has led to worsening air quality in many of these densely populated regions. This is driven by the increasing emissions of nitrogen oxides ($NO_x = NO + NO_2$), largely associated with transport, and volatile organic compounds (VOCs), released from a diverse range

of sources. Photochemical reactions in the atmosphere then lead to the formation of a wide range of important secondary pollutants, including ozone ($O_3$) and secondary inorganic and organic aerosol.

Tropospheric $O_3$ is both an air pollutant and an important greenhouse gas throughout the troposphere (Skeie et al., 2020). High levels of $O_3$ can adversely affect vegetation, global crop yields (Avnery et al., 2011) and human health, with long-term exposure

increasing the risk of death from cardiovascular and respiratory illnesses (Jerrett et al., 2009), and short-term exposure leading to the exacerbation of asthma in children (Thurston et al., 1997). $O_3$ exposure has been linked to both acute and chronic pulmonary and cardiovascular health outcomes through both animal toxicological and human clinical studies, with one study showing statistically significant decreases in the lung function of adults on an average exposure of 70 ppbV of $O_3$ across five 6.6-hour windows (WHO 2005 and 2013, Schelegle et al., 2009, EPA 2013, Fleming et al., 2018).


As a result of increased anthropogenic emissions, tropospheric $O_3$ increased globally during the 20[th] century, and has continued to rise regionally in Asia during the 21[st] century (Fleming et al., 2018; Royal Society, 2008, Lu et al., 2020). Background tropospheric $O_3$ has also continued to increase (Parrish et al., 2014, Tarasick et al., 2019). Since the 1990s, surface $O_3$ trends have varied by region (Gaudel et al., 2018, Cooper et al., 2020, Lu et al., 2020), but trends in the free troposphere have been

overwhelmingly positive (Gaudel et al., 2020, Liao et al., 2020). Both satellite data and global chemical transport models have identified India and East Asia as the region with the greatest $O_3$ increases between 1980-2016 (Ziemke et al., 2019), with the rate of change per decade between 2005-2016 more than double that of the rate between 1979-2005 (Ziemke et al., 2019).

Unlike other pollutants such as $NO_x$ and $SO_2$, ground-level $O_3$ is not directly emitted but is formed in the atmosphere from the photochemical processing of a range of reactive precursor species (Calvert et al., 2015). $O_3$ reduction strategies are complicated by its non-linear relationship with its precursor species $NO_x$, CO and VOCs; their reduction does not necessarily lead to a reduction in $O_3$, and $O_3$ production is also influenced by longer lived gases such as $CH_4$ and CO. The urban atmosphere can be classified as being either VOC-limited or $NO_x$-limited. In a VOC-limited environment, $O_3$ can be effectively reduced by reducing VOCs, whereas decreasing $NO_x$ may have a limited effect or even increase the local $O_3$ production rate, $P(O_3)$ (Li et al., 2021). In a $NO_x$-limited regime, reducing $NO_x$ emissions is the most efficient approach for reducing $O_3$ production. Before implementing emission abatement strategies, it is therefore important to consider which regime prevails. Highly populated urban areas are commonly VOC-limited, meaning reducing $NO_x$ without also reducing VOC sources may potentially lead to an increase in local $P(O_3)$ (Sillman, 1999; Sillman et al., 1990) and hence lead to an increase in ground-level $O_3$ concentrations.

In general, $O_3$ formation is mediated by the reactions of peroxy radicals, $RO_2$ and $HO_2$, formed in the OH-initiated oxidation of VOCs (R1), with NO to produce $NO_2$ (R2 and R4). $NO_2$ is then rapidly photolyzed back to NO, forming $O(^3P)$ (R5), which can rapidly react with $O_2$, leading to $O_3$ (R6). This recycling of NO to $NO_2$ leads to net production of $O_3$ (Calvert et al., 2015). A description of how net $O_3$ production, $P(O_3)$, is calculated from observed and modelled concentrations can be found in section 3.4.

$$RH + OH + O_2 \rightarrow RO_2 + H_2O \qquad \text{(R1)}$$
$$RO_2 + NO \rightarrow RO + NO_2 \qquad \text{(R2)}$$
$$RO_2 + O_2 \rightarrow R´{=}O + HO_2 \qquad \text{(R3)}$$
$$HO_2 + NO \rightarrow OH + NO_2 \qquad \text{(R4)}$$

$$NO_2 + h\nu \rightarrow NO + O(^3P) \quad (\lambda > 420 \text{ nm}) \qquad \text{(R5)}$$
$$O(^3P) + O_2 (+M) \rightarrow O_3 (+M) \qquad \text{(R6)}$$

The propensity of a particular VOC species to enhance $O_3$ production is determined by its reactivity, structure and ambient concentration (Jenkin et al., 2017). The sources of emissions, and their relative magnitudes, differ between cities, leading to differences in the role played by individual VOCs with respect to $O_3$ production. Previous studies in megacities have found that a range of different VOC classes lead to ground-level $O_3$ production. A comprehensive study of chemical processing in London, during the 2012 summer ClearfLo campaign, found biogenic and longer-chain VOCs from diesel sources to be of greatest importance to VOC-hydroxyl radical reactivity, and thus in situ $O_3$ formation (Dunmore et al., 2015; Whalley et al., 2016). Studies investigating the sensitivity of $O_3$ production to different VOC classes in Shanghai (2006 – 2007) found that monoaromatic species dominated, accounting for 45 % of the total $O_3$ production (Geng et al., 2008). Another study in Shanghai (summer 2009) identified significant contributions to $O_3$ production from but-2-enes (Ran et al., 2011). Biogenic

species, such as isoprene, were found to have the greatest impact on OH reactivity, $k$(OH), an indicator for $O_3$ production, in Seoul in 2015 (Kim et al., 2016) and the Pearl River Delta in 2006 (Lou et al., 2010), whereas another study pointed to the importance of monoaromatics, followed by isoprene and anthropogenic alkenes to $O_3$ production in Seoul in spring 2016 (Simpson et al., 2020). These studies incorporated a variety of chemical detail into their models, dependent on the breadth of their VOC measurement suite, with many studies not accounting for oxygenated, or biogenic species other than isoprene (Lou et al., 2010). Zavala et al., 2020 show that the key contributors to VOC-OH reactivity in the Mexico City Metropolitan Area (MCMA) between the 1990s and 2019 have changed, with aromatic and alkene contributions decreasing with reduced VOC emissions from mobile source. Oxygenated VOCs from solvent consumption and personal care products dominate the VOC-OH reactivity in recent years, leading to sustained high $O_3$ concentrations in the MCMA (McDonald et al., 2018, Zavala et al., 2020). Understanding which precursor species are key to $O_3$ production in any given city allows governments to introduce measures to combat air quality problems (Molina, 2021).

Over the past two decades, advances in vehicle emission technology, along with improvements in residential heating, have led to decreased $NO_x$ emissions in industrialised and highly populated regions of the western world (Georgoulias et al., 2019). In many European cities, additional measures banning vehicle types in busy areas during weekdays, and upgrading the Heavy Goods Vehicle (HGV) vehicle fleet, has led to further reductions in ambient $NO_2$ (Font et al., 2019). In 2013, China introduced the Air Pollution Prevention and Control Action Plan. (Zhang et al., 2019). New measures included the improvement of industrial emissions standards, the promotion of cleaner fuels to replace coal in the residential sector, and the removal of older vehicles from the roads. As a result of these new controls, $NO_2$ emissions in Beijing have decreased by 32 % since 2012 (Krotkov et al., 2016; Liu et al., 2016; Miyazaki et al., 2017). Despite these successes, surface $O_3$ pollution across China has continued to increase (Li et al., 2019b; Lu et al., 2018a). The overall $O_3$ formation potential (OFP) of VOCs has increased alongside this, despite reduction strategies leading to reduced emissions of alkanes and alkenes. This is explained by an increasing emission of VOC species with higher OFPs, such as toluene and xylenes, driven by solvent use and industrial processes (Li et al., 2019b).

The megacity of Delhi, with an estimated population of over 28 million in 2018 (United Nations, 2019), suffers from some of the world's poorest air quality (World Health Organization, 2014). Significant efforts have been made over the past two decades to reduce the air pollution burden in Delhi, including the introduction of fuel quality standards, a new metro system to improve public transport, and an odd-even traffic number plate system (Bansal and Bandivadekar, 2013; Kumar et al., 2017). Since the late 1980s, steps to mitigate the impacts of vehicle and fuel emissions in India have been implemented. Initial interventions included switching to compressed natural gas (CNGs) for autorickshaws and buses in Delhi and other major cities. Since 2010, Bharat IV fuel quality standards have been implemented across cities in India, based on the 2003 Auto Fuel Policy. These changes have resulted in reduced annual $NO_x$ emissions nationally, compared with the projected emissions without policy implementation (Bansal and Bandivadekar, 2013).

A recent study of the implications of the SARS-CoV-2 pandemic on air quality compared pollutant levels of $PM_{2.5}$, $PM_{10}$, $NO_2$, CO and $O_3$ in Delhi, and across other Indian cities, before and after a national lockdown (Jain and Sharma, 2020; Mahato et al., 2020; Sharma et al., 2020). The results showed significant improvements in post lockdown air quality, with reductions in ambient $NO_2$, $PM_{2.5}$ and $PM_{10}$ exceeding 50 % compared to business as usual. However, increased concentrations of ground-level $O_3$ (> 10 %) were also observed and were attributed to reductions of NO leading to reduced consumption of $O_3$. Another study found that, after removing meteorological biases, concentrations of $NO_2$ and $PM_{2.5}$ at urban background sites in Delhi to have reduced by ~ 51 % and ~ 5 % respectively, with $O_3$ concentrations increasing by ~ 8 % (Shi et al., 2021). Prior to the SARS-CoV-2 pandemic, studies have observed high concentrations of ambient $O_3$ in New Delhi, with high levels associated with anti-cyclonic conditions (sunny and warm, stagnant winds and lower humidity) typical during October, the month in which the observational dataset used in this study was obtained (Jain et al., 2005; Sharma et al., 2016). Poor air quality is exacerbated in late October - November, when additional $O_3$ precursor pollutants are emitted from regional agricultural biomass burning of crop residues within the wider area, and from firecrackers and the burning of effigies as part of seasonal festivities (Jain et al., 2014; Sawlani et al., 2019).

Several recent studies have examined the sources of VOCs in Delhi. Top down and bottom-up approaches have shown gas-phase organic air pollution to be predominantly from petrol (gasoline) and diesel fuel sources. In two 2018 studies at two different urban sites in Delhi, Stewart et al., 2021a and Wang et al., 2020 showed that 52 % and 57 % of the measured mixing ratio were from combined petrol and diesel sources. Smaller contributions to the overall VOC burden were found from solid fuel combustion, 16 % (Stewart et al., 2021a) and 27 % (Wang et al., 2020). These data were in line with Guttikunda and Calori, 2013, who produced an inventory which showed that petrol and diesel sources were responsible for 65% by mass of hydrocarbons in Delhi. VOC emissions have also been shown to be dominated by petrol and diesel sources (~ 50 %) in a study which conducted positive matrix factorisation (PMF) analysis on proton transfer reaction mass spectrometer flux measurements taken in a follow-on campaign in early November 2018 at IGDTUW (Cash et al., 2021, in preparation).

There have been several recent studies focussed on understanding VOC emissions from sources specific to Delhi. Stewart et al., 2021b studied the types of intermediate-volatility and semi-volatile gases released from solid fuels in Delhi to better understand their potential impact on air quality, and Stewart et al., 2021c produced highly speciated non-methane VOC emission factors from a range of solid fuel combustion sources characteristic to Delhi, which were used by Stewart et al., 2021d for use in regional policy models and global chemical transport models. Stewart et al., 2021e found that fuel wood, crop residue, cow dung cake and municipal solid waste burning were shown to be 30, 90, 120 and 230 times more reactive with the OH radical, which can lead to $O_3$ formation, than liquified petroleum gas (LPG), and may be one the factors for the high $O_3$ levels, and overall poor air quality, observed.

To implement successful ground-level $O_3$ reduction strategies, a good understanding of the non-linear, chemically complex processing of its precursor species is imperative. This paper presents a comprehensive in situ $O_3$ production sensitivity study, using an extensive speciated VOC and o-VOC measurement suite obtained in Delhi during the Atmospheric Pollution and Human Health program in an Indian Megacity (APHH-India) DelhiFlux project post-monsoon field campaign in October 2018. Measurements of $NO_x$, CO, $SO_2$, $O_3$, HONO, 34 photolysis rates, pressure, temperature and relative humidity complete the data set. The observations were used to constrain a detailed zero-dimensional chemical box model, incorporating the Master Chemical Mechanism v3.1.1 (MCM: mcm.york.ac.uk). Significant VOC classes and sources contributing to the OH reactivity ($k$(OH)) are identified by constraining the model to the full VOC suite. By modifying the constraints of these species, their contribution to in situ $O_3$ production was investigated by comparing relative changes in the modelled rate of $O_3$ production. The sensitivity of in situ $O_3$ production to changes in NO, photolysis and PM was also investigated.

## 2 Experimental

### 2.1 Site description

The APHH-India DelhiFlux post-monsoon measurement campaign took place between 4th October and 5th November 2018. The primary measurement site was located on the campus of the Indira Gandhi Delhi Technical University for Women (IGDTUW), near Kashmiri gate in Old Delhi (28.67° N, 77.23° E). The campus is an open area, with some green spaces, and is close to major roads and highways: 1.5 km north of the busy Chandni Chowk market, 0.6 km north of Old Delhi (Delhi Junction) Railway Station and 0.3 km west of the National Highway 44 (Figure 1). The campus is mostly pedestrianised, with occasional traffic activity from delivery cars, taxis and autorickshaws. Inter State Bus Terminal (ISBT) is located < 100 m away from the measurement site. IGDTUW facilitated the sampling of ambient air from a height of ~ 5 m and measurements were made of a large range of VOCs, o-VOCs, $NO_x$, CO, $SO_2$, HONO, photolysis rates and PM.

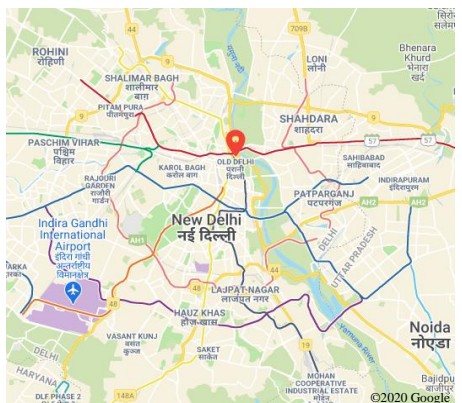
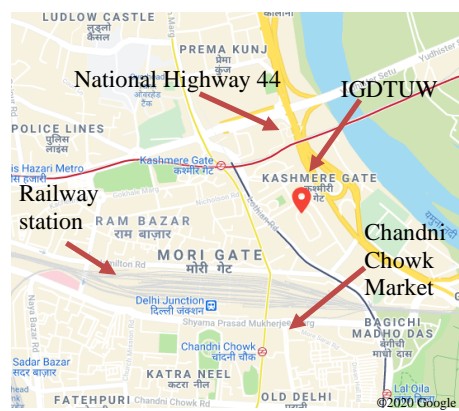

**Figure 1:** The measurement site (IGDTUW) in Old Delhi, north of New Delhi (left). In closer detail (right), the site was just north of Delhi Junction Railway Station, and west of National Highway 44. Map data ©2020 Google.

## 2.2 VOC and o-VOC measurements

Two gas chromatography (GC) instruments and a proton transfer reaction time-of-flight mass spectrometer, with quadrupole ion guide (PTR-QiTOF, Ionicon Analytik, Innsbruck) were deployed to measure an extensive range of VOC and oxygenated VOCs (o-VOCs). GC instrumentation included a dual-channel gas chromatography flame ionisation detector, (DC)-GC-FID, measuring $C_2$-$C_8$ hydrocarbons and some o-VOCs and a two-dimensional GC flame ionisation detector (GC × GC-FID) to measure larger hydrocarbon species ($C_6$-$C_{13}$). PTR-QiTOF measurements of a variety of o-VOCs completed the VOC measurement suite of 15 alkanes, 10 alkenes, 2 alkynes, 29 aromatics, 11 carbonyls, 2 alcohols and 15 monoterpenes (see table S1 in supplementary). All three instruments operated between 11[th] October and 27[th] October 2018. The two GC instruments shared an inlet, located approximately 5 m above ground level. The sample line from the inlet to the laboratory was made from ½" O.D. (9.53 mm I.D.) perfluoroalkoxy (PFA) and was heated to limit adsorption of compounds to surfaces. VOCs and o-VOCs were calibrated using 4 ppbv and a 4 ppmv (diluted before calibration) gas standard cylinders (National Physical Laboratory, UK) respectively, containing a variety of VOCs and o-VOCs. The linearity of the detector response to higher mixing ratios was tested on both instruments prior to the campaigns, as described by Stewart et al., 2021a.

The (DC)-GC-FID measurement period was 5[th] October – 27[th] October 2018. A 500 mL sample pre-purge (flow rate of 100 mL min[-1] for 15 mins) preceded a 500 mL sample collection (flow rate of 25 mL min[-1] for 20 mins) using a Markes International CIA Advantage. The sample was passed through a cold glass finger (- 30 °C) to remove water, before being adsorbed onto a Markes International Ozone Precursor dual-bed sorbent cold trap using a Markes International Unity 2 for pre-concentration. After sampling, the trap was heated (250 °C for 3 mins) to allow for thermal desorption of the sample and passed to the GC oven in helium carrier gas. The sample was split 50:50 and injected into two columns in the oven, allowing for the detection of both non-oxygenated (10 m x 0.53 mm LOWOX column) and oxygenated (50 m x 0.53 mm $Al_2O_3$ PLOT column) VOCs. The oven was held at 40 °C for 5 mins, then ramped up to 110 °C at a rate of 13 °C min[-1], before a final ramp to 200 °C at 7 °C min[-1] where it was held for 30 mins (Hopkins et al., 2003).

The GC × GC-FID measurement period was 11[th] October – 4[th] November 2018. A 2.1 L sample (flow rate 70 mL min[-1] for 30 mins) was collected and passed through a cold glass finger (-30 °C) to remove water. The sample was absorbed onto a TO-15/TO-17 air toxics cold sorbent trap in a Markes International Unity 2 for pre-concentration. The trap was heated (250 °C for 5 mins) to allow for thermal desorption and the sample injected down a transfer line. It was then refocused with liquid $CO_2$ at the head of a non-polar BPX5 column (SGE Analytical, 15m x 0.15 μm x 0.25 mm) held at 50 psi for 60 seconds. This was connected to a polar BPX50 column (SGE Analytical 2 m x 0.25 μm x 0.25 mm) held at 23 psi via a modulator held at 180 °C (5 s modulation, Analytical Flow Products MDVG-HT). The oven was held for 2 mins at 35 °C, then ramped up to 130 °C at a rate of 2.5 °C min[-1] and held for 1 min before a final ramp to 180 °C at 10 °C min[-1] where it was held for 8 mins (Dunmore et al., 2015; Stewart et al., 2021a).

The PTR-QiTOF-MS operated from 4$^{th}$ October – 4$^{th}$ November 2018 at a flow rate of 20 L min$^{-1}$, subsampling from the same inlet line as the GCs. The drift tube pressure and temperature were 3.5 mbar and 60 °C, respectively, giving an *E/N* (the ratio between electric field strength, *E*, and buffer gas density, *N*, in the drift tube) of 120 Td. The PTR-QiTOF-MS subsampled from a ½ inch PFA inlet line positioned 5m above ground next to the inlet used by the GC instruments. The PTR-QiTOF-MS was calibrated daily using a calibration gas (Apel-Riemer Environmental Inc., Miami) containing 18 compounds: methanol; acetonitrile; acetaldehyde; acetone; dimethyl sulphide; isoprene; methacrolein; methyl vinyl ketone; 2-butanol; benzene; toluene; 2-hexanone; m-xylene; heptanal; α-pinene; 3-octanone and 3-octanol at 1000 ppbv (±5 %); and β-caryophyllene at 500 ppbv (±5 %), diluted dynamically into zero air. More details on the PTR-QiTOF-MS can be found in Jordan et al., 2009.

## 2.3 Measurements of NO$_x$, CO, SO$_2$, O$_3$ and HONO

Measurements of nitrogen oxides (NO$_x$) were made using a dual channel chemiluminescence instrument (Air Quality Designs Inc., Colorado) and carbon monoxide (CO) was measured with a resonance fluorescent instrument (Model Al5002, Aerolaser GmbH, Germany). Both instruments were calibrated every 2-3 days throughout the campaign using standard gas cylinders from the National Physical Laboratories, UK. O$_3$ was measured using an ozone analyser (49i, Thermo Scientific) with a limit of detection of 1 ppbv. The instrument setup and calibration methodology are as described by Squires et al., 2020.  An SO$_2$ analyser (43i, Thermo Scientific) with a limit of detection of 2 ppbv was used to provide measurements of SO$_2$.  HONO was measured with a commercially available long path absorption photometer (LOPAP) described in Heland et al., 2001 according to the standard procedures outlined in Kleffmann and Wiesen, 2008, with baseline measurements taken at regular intervals (8 h). In this study HONO was also recorded using the PTR-QiTOF-MS with the protonated HONO ion observed at m/z 48.007. This signal was humidity corrected and calibrated against the LOPAP HONO measurements. HONO measurements using PTR-MS have been reported previously Koss et al., 2018, and are described in more detail in Crilley et al., 2021, in preparation.

## 2.4 Measurement of photolysis rates

The model was constrained with the measured photolysis frequencies *j*(O1D), *j*(NO$_2$) and *j*(HONO)), which were calculated from the measured wavelength-resolved actinic flux and published absorption cross sections and photodissociation quantum yields (Whalley et al., 2020). All other photolysis rates are parameterised as a function of solar zenith angle using a two-stream isotropic scattering model as described by Jenkin et al., 1997 and Saunders et al., 2003. In each case, clear sky variation of a specific photolysis rate (*j*) with solar zenith angle ($\chi$) can be described well by the expression 1:

$$j = l \cos(\chi) m \times e -n \sec(\chi) \tag{1}$$

where the parameters *l*, *m* and *n* optimised for each photolysis rate (see Table 2 in Saunders et al., 2003 and http://mcm.york.ac.uk/parameters/photolysis_param.htt)

For species which photolyse at near-UV wavelengths (< 360 nm), such as HCHO and $CH_3CHO$, the photolysis rates were calculated by scaling to the ratio of clear-sky $j(O^1D)$ to observed $j(O^1D)$ to account for attenuation by clouds and aerosol. For
species which photolyse further into the visible, the ratio of clear-sky $j(NO_2)$ to observed $j(NO_2)$ was used.

### 2.5 Model description

A tailored zero-dimensional chemical box model of the lower atmosphere, incorporating a subset of the Master Chemical Mechanism (MCM v3.3.1; Jenkin et al., 2015) into the AtChem2 modelling toolkit (Sommariva et al., 2020), was used to identify the main drivers of in situ $O_3$ production, $P(O_3)$, in Delhi. The MCM describes the detailed atmospheric chemical
degradation of 143 VOCs, though 17,500 reactions of 6,900 species. More details can be found on the MCM website (http://mcm.york.ac.uk). Observations of 86 unique VOCs, $NO_x$, CO, $SO_2$ and total aerosol surface area (ASA), along with 34 observationally derived photolysis rates, temperature, pressure and relatively humidity, were averaged or interpolated to 15-minute data and used to constrain the model. Some measured VOCs are not described in the MCM. For these species, an existing mechanism in the MCM was used as a surrogate mechanism. Surrogate species were selected based on their structural
similarly to the species of interest, and reaction rates with OH, $HO_2$ and $NO_3$ were amended to values found in the IUPAC atmospheric chemical kinetics database (www.iupac.pole-ether.fr) and Atkinson and Arey, 2003 (see table S1). A fixed deposition rate of $1.2 \times 10^{-5}$ $s^{-1}$ was used, giving model generated species a lifetime of *ca.* 24 hours.

The model was constrained to an adjusted value of the observed HONO to account for high surface concentrations and an
expected decline in concentration with height. The rate of vertical transport of chemical species through turbulence, or Deardorff velocity ($w^*$), was calculated during a 2-week follow-on chemical flux campaign in early November 2018 at IGDTUW, after the ground level measurement period ended. Measurements made from a 30 m tower allowed for the calculation of $w^*$, from which an hourly averaged diel has been used to determine an approximate rate of vertical HONO transport. Observed HONO concentrations ( $[HONO]_{meas}$ ) were converted to an adjusted HONO concentration
($[HONO]_{input}$) with equation 2:

$$[HONO]_{input} = [HONO]_{meas} \, e^{-j(HONO)t} \tag{2}$$

where *t* is calculated as the time for $[HONO]_{meas}$ to travel to half the boundary layer height at the measured Deardorff velocity, $w^*$. The first order loss of $HO_2$ to aerosol surface area ($k$) was calculated with equation 3.

$$k = \frac{\omega A \gamma}{4} \tag{3}$$


where $\omega$ is the mean molecular speed of $HO_2$ (equal to 43725 cm s$^{-1}$ at 298 K), $\gamma$ is the aerosol uptake coefficient (0.2 is used here as recommended by Jacob, 2000), and $A$ is the measured aerosol surface area in cm$^2$cm$^{-3}$ calculated (as NSD.$\pi$.d$^2$) from hourly number size distributions (NSD). Aerodynamic diameter NSD, $N_a$ (> 352 nm) were collected using a GRIMM 1.108 (Portable Laser Aerosol Spectrometer and Dust Monitor) and merged into concurrently measured Electric mobility diameters

NSD, $N_d$ (14 – 640 nm – measured using a TSI SMPS, consisting of a 3080 Electrostatic Classifier, 3081 DMA and 3775 CPC) using the algorithm developed by Beddows et al., 2010.

The base reference model was run for 5 days, with each day being constrained to the diel campaign averaged observations, to allow for the spin-up of model generated species. Only the fifth day was taken for analysis to ensure steady state was reached.

A range of additional modelling scenarios, based on this base model, were used to investigate the sensitivity of $O_3$ production, $P(O_3)$, to changes in VOCs, $NO_x$, aerosol surface area and/or photolysis rates:

- Scenario 1: Vary both total VOC and $NO_x$ by a factor. VOC and $NO_x$ observations were multiplied by a factor of 0.01, 0.1, 0.25, 0.5, 0.75, 0.9, 1.1, 1.25, 1.5, 1.75 and 2.

- Scenario 2: Vary individual primary VOCs and $NO_x$ by a factor. A VOC of interest and $NO_x$ observations were

multiplied by a factor of 0.01, 0.1, 0.25, 0.5, 0.75, 0.9, 1.1, 1.25, 1.5, 1.75 and 2. Observed carbonyls (all oxygenated compounds excluding alcohols) were not constrained in this scenario to allow for secondary compounds to be varied with changes in concentrations of their precursor VOC.

- Scenario 3: Vary by VOC class and vary $NO_x$. Individual VOC classes were multiplied by a factor of 0.2, 0.4, 0.6, 0.8, 1.2, 1.4, 1.6, 1.8 and 2. $NO_x$ was multiplied by a factor of 0.25, 0.5 and 0.75. All possible combinations of altered

VOC class and $NO_x$ observations were constrained in the model. The remaining VOCs not grouped into the class of interest are constrained at their observed values. Observed carbonyls are not constrained in this scenario to allow for secondary compounds to be varied with changes in concentrations of their precursor VOC.

- Scenario 4: Increase individual VOC of interest by 5 %. The constrained concentration of a VOC species of interest was increased by 5 % by molar mass following the procedure carried out in Elshorbany et al., 2009. Comparison of

the change in $P(O_3)$ against the base model allows for the determination of $\Delta P(O_3)_{increm}$ (section 3.4). Observed carbonyls are not constrained in this scenario to allow for secondary compounds to be varied with changes in concentrations of their precursor VOC.

- Scenario 5: Vary total aerosol surface area (ASA). ASA was multiplied by a factor of 0.1, 0.3, 0.5, 0.7, 0.9, 1.1, 1.3, 1.5, 1.7, 1.9 and 2 The difference in $P(O_3)$ between each model and the base model was examined.

• Scenario 6: Vary photolysis rates. All 34 photolysis rates were multiplied by a factor of 0.1, 0.3, 0.5, 0.7, 0.9, 1.1, 1.3, 1.5, 1.7, 1.9 and 2. The difference in $P(O_3)$ between each model and the base model was examined.

• Scenario 7: Vary both individual primary VOCs and $NO_x$ by EDGAR source sector. VOCs were grouped and observed concentrations were divided into each source sector as described by the EDGAR v5.0 Global Air Pollutant Emissions and EDGAR v4.3.2_VOC_spec inventories (https://edgar.jrc.ec.europa.eu/#). The proportion of VOC and 330 $NO_x$ concentrations attributed to each individual source were multiplied by a factor of 0.01, 0.1, 0.25, 0.5, 0.75, and 0.9. Observed carbonyls were not constrained in this scenario to allow for secondary production of these compounds to be varied with changes in concentrations of their precursor VOC. For each of these model runs, there were 6 variations where monoterpenes were assumed to be 0 %, 10 %, 25 %, 50 %, 75 % and 100 % anthropogenic.

**3 Results**

**3.1 Observed $NO_x$, CO and $O_3$**

The observed mixing ratio (ppbv) timeseries of NO, $NO_2$, CO and $O_3$ during the campaign are presented in Figure 2. High concentrations of NO and CO were observed towards the latter half of the month, with high day time $O_3$ concentrations observed throughout the entire measurement period.


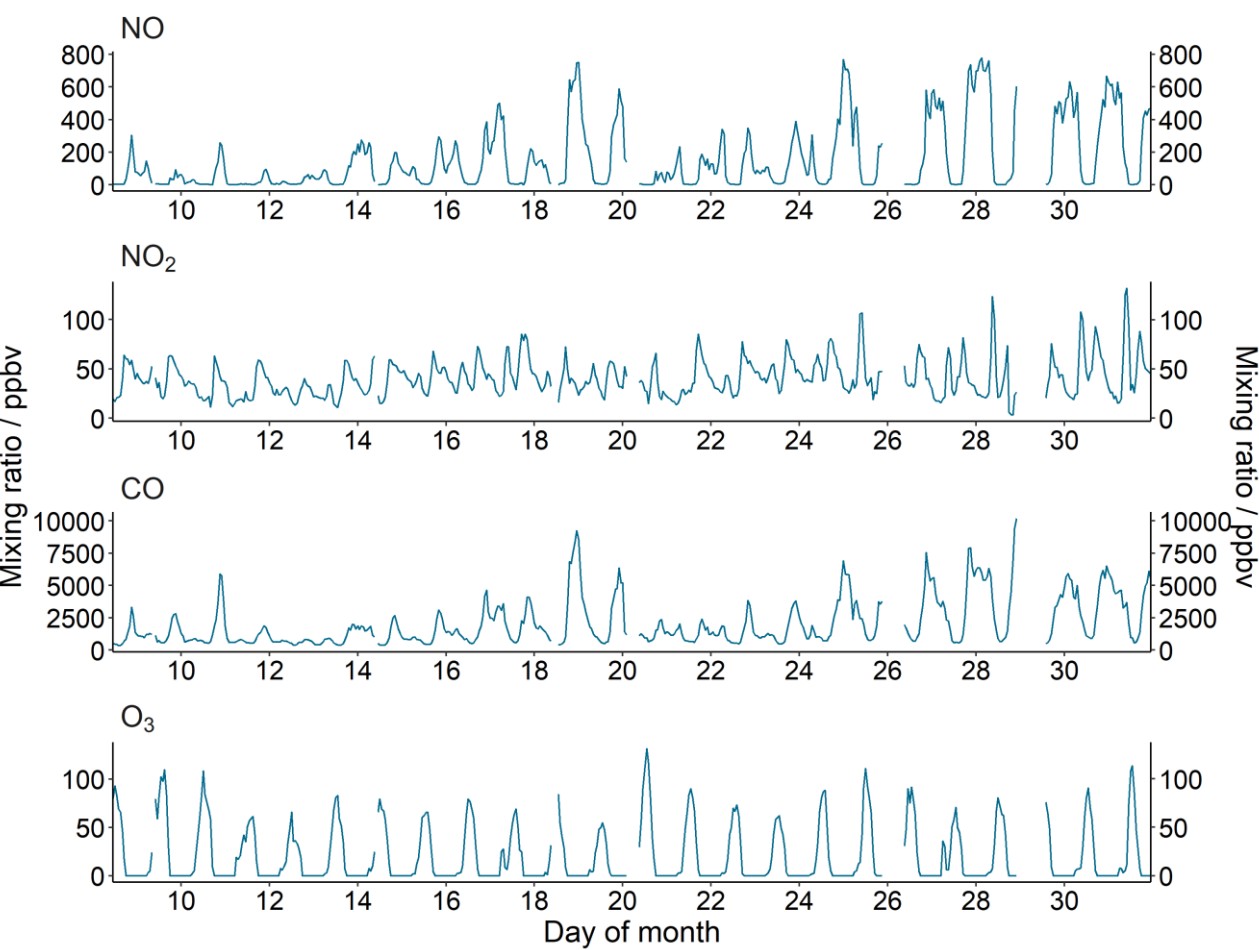

**Figure 2:** Observed mixing ratio timeseries of (from top to bottom): NO, $NO_2$ CO and $O_3$ during October 2018.

The diel profile of NO, and to a lesser extent CO, have a U-shaped profile, with much higher concentrations observed at night compared to the day (Figure 3). This profile results from a shallow and stagnant nocturnal boundary layer, described in more detail in Stewart et al., 2021a. Average daytime concentrations (06:00 – 18:00, roughly in-line with sunrise and sunset times) of NO and CO were 58.8 ppbv and 1.2 ppmv respectively. These campaign averaged diel profiles have large standard deviations, representing the day-to-day variation in concentration throughout October (Figure 2). Average night-time concentrations (18:00-06:00) of NO and CO were 247.0 ppbv and 2.9 ppmv respectively, with NO and CO concentrations occasionally exceeding 800 ppbv and 10 ppmv respectively in the latter half of the month (Figure 2). The $NO_2$ profile shows two peaks, at approximately 09:00, and 18:00, perhaps due to increased commuting traffic during these times. The $O_3$ diel profile peaks at approximately 13:00, with a mean peak concentration of 78.3 ppbv. A rapid increase in $O_3$ concentration is observed first thing in the morning from around 05:00, which levels off at around 06:00 before again rapidly increasing from

08:00 to a peak concentration ~ 13:00. As the precursor species of $O_3$ depend on light to undergo the chemical processing that
leads to $O_3$ production, this profile suggests that as the sun rises rapid photochemical formation of $O_3$ is initiated by the photolysis of precursors that have accumulated in the lower atmosphere during the night-time. The early morning increase in $O_3$ concentrations may also be influenced by the reduced NO titration as the boundary layer height increases. (Figure 3).

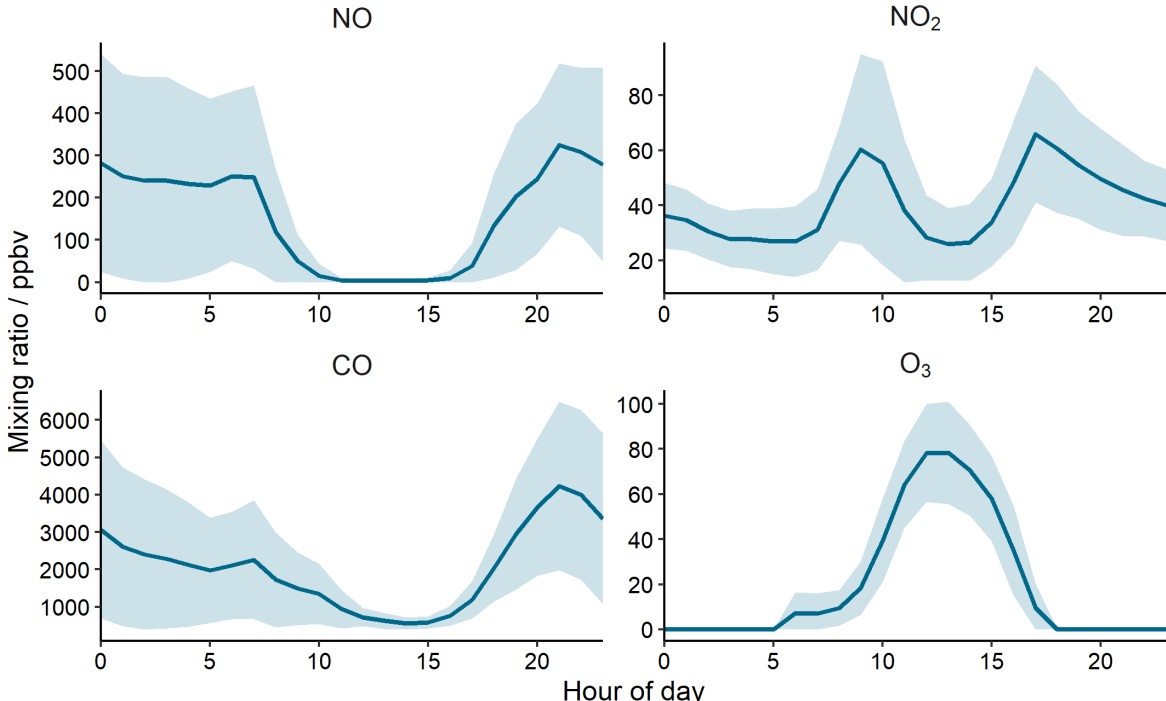

**Figure 3:** October campaign averaged diel profiles. The shaded ribbon represents the standard deviation of this average.


Average mixing ratios of $NO_x$ and VOCs observed during the post-monsoon Delhi campaign are comparable to those observed in 1970s Los Angeles. High concentrations of $NO_x$ ($\approx$ 200 ppbV) and VOCs (e.g. benzene $\approx$ 10 ppbV; toluene $\approx$ 30 ppbV) led to large concentrations of $O_3$ in the US city, averaging at around 400 ppbV (Pollack et al., 2013). Despite the high $O_3$ concentrations observed in Delhi, $O_3$ concentrations in post-monsoon Delhi did not exceed 150 ppbV and are of similar
magnitude to observed $O_3$ in Beijing, Shanghai and Guangzhou despite much higher observed VOC and $NO_x$ concentrations (Tan et al., 2019). The lower $O_3$ observed in Delhi compared to 1970s Los Angeles can be attributed to differences in both topography and meteorology. The isolated coastal city of Los Angeles is surrounded by mountains to the North and East, with the prevailing wind dominating from the coast. Owing to its basin-like topography, and with a cool on-shore sea breeze often creating a temperature inversion, the air mass circulates within the city and the transport of emissions out of the basin is
impeded. Although landlocked Delhi lies to the southwest of the Himalayas, the city is very flat and resides far enough away from the mountain range to allow for the efficient transportation of air masses from the city. It is also important to consider

that the very high concentrations of $NO_x$ and VOCs observed peak to comparable concentrations to Los Angeles during the evening and at night, where they are trapped owing to a shallow, stagnant boundary layer, and there is little to no photochemical activity. $O_3$ production rates in Delhi peaks in the morning, when concentrations of pollutants, though still high, are much lower than at night due to rapid boundary layer expansion (see section 3.4).

Although twelve key air pollutants, including $O_3$, have prescribed national ambient air quality standards (Dube, 2009), only four have been identified for regular and continuous monitoring: sulfur dioxide ($SO_2$), nitrogen dioxide ($NO_2$), respirable suspended particulate matter (RSPM, or $PM_{10}$), and fine particulate matter ($PM_{2.5}$). The prescribed national standards for $O_3$ are an 8-hourly limit of 100 µg m$^{-3}$ (~ 50 ppbv), and 1-hourly limit of 180 µg m$^{-3}$ (~ 90 ppbv). The observed $O_3$ 8-hour averages between 09:00 and 17:00 throughout the measurement period are shown in Figure 4, along with the hourly averaged maximum $O_3$ concentrations.

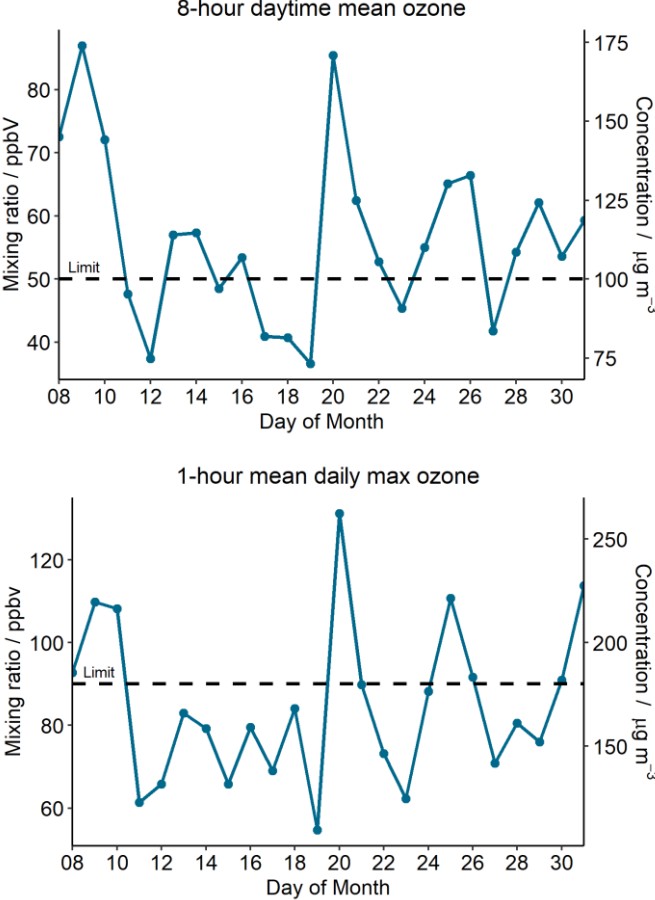

**Figure 4:** 8-hour mean $O_3$ concentration between 09:00 and 17:00 IST (top) and hourly averaged maximum daily $O_3$ concentration (bottom) during October 2018. The black dashed line is the prescribed national standards for $O_3$.

Our observations show that the 8-hour $O_3$ prescribed national standard is exceeded on 16 days during our 24-day measurement period (67 % of days), and the 1-hour max is exceeded on 8 days (33 % of days). The published national standards state that any pollutant which exceeds the prescribed values for two consecutive days qualifies for regular and continuous monitoring. As there are up to four consecutive days in which the 8-hour daytime mean $O_3$ limit was breached during our campaign, this implies that $O_3$ should be continuously and regularly monitored in this part of Old Delhi.

## 3.2 Observed volatile organic compounds (VOCs)

Alkanes were the predominant VOC class contributing to the total measured VOC mass concentration (42 %), consistent with previous observations at the site (Shivani et al., 2018), followed by alcohols (18 %), aromatics (17.8 %) and carbonyls (13.9 %). The percentage contributions of each VOC class to total measured VOCs, along with mean mass concentrations of the top 10 contributors, are presented in figure 5. The top individual species contributing to the VOC mass concentrations were ethanol (10.7 %), *n*-butane (9.9 %) and *n*-propane (7.9 %).

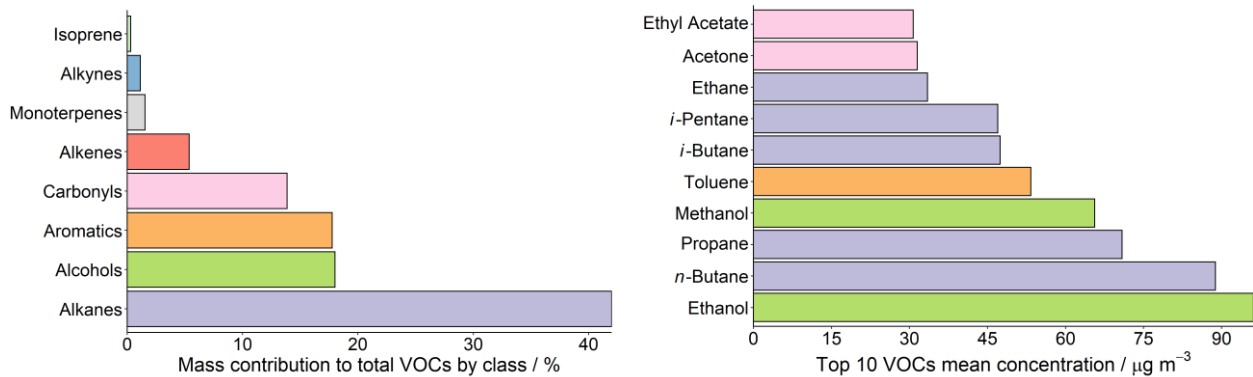

**Figure 5**: Percentage contribution of different VOC classes to the total mean measured NMVOCs (left) and the mean concentrations of the top 10 contributors to total measured VOC concentrations (right) during the campaign in µg m$^{-3}$. Colours correspond to the different VOC classes.

The general diel profile for all VOCs, excluding isoprene and some oxygenated species, is U-shaped (figure 6). This U-shaped profile is also observed for NO and CO. Concentrations are much higher during the night, and lower in the day, as they are concentrated in a stagnant, shallow boundary layer that forms over the city at night (Stewart et al., 2021a), and are subject to photochemical losses during the day. The U-shape is less apparent for acetone, methanol and ethanol. This may be a result of very high emissions of these compounds during the day and/or formation through secondary chemistry (Stewart et al., 2021a).

Isoprene has a typical biogenic diurnal profile, in contrast to monoterpenes which have a similar trend to other anthropogenic species (see α-Terpinene, Figure 6). A separate study which conducted Positive Matrix factorisation (PMF) analysis on the co-located PTR-QiTOF flux measurements, resolved two traffic related factors (Cash et al., 2021). A large proportion of the total
measured monoterpenes were resolved within traffic factors (~ 60 %) and traffic emissions dominated throughout the campaign (~50 % of total VOC emissions). Gkatzelis et al., 2021 have recently shown that monoterpene measurements in New York City can be predominantly apportioned to fragranced volatile consumer products (VCPs) and other anthropogenic sources (e.g., cooking and building materials), and that although biogenic sources are the dominant source of monoterpenes globally, in urban environments monoterpene fluxes from fragranced VCPs can compete with the emissions from local vegetation (see
Section 3.8 for further discussion of anthropogenic sources of monoterpenes).

The standard deviation of all the aggregate VOC diel profiles is large, owing to large variations in concentrations day to day throughout the month. Generally, high concentrations of all species are observed at the end of October, and lower concentrations at the start (see supplement Figures S1 and S2).

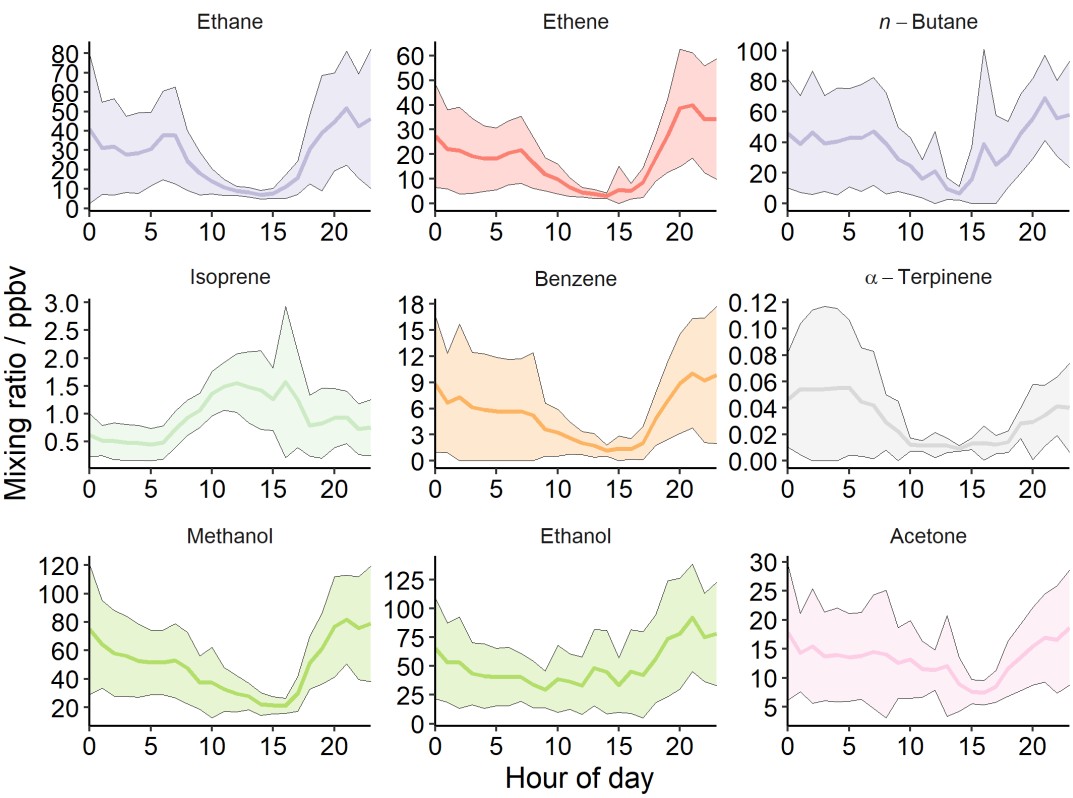


**Figure 6**: Mean campaign averaged diel mixing ratio profiles of selected VOCs during the campaign: ethane, ethene, n-butane, isoprene, benzene, α-terpinene, methanol, ethanol, acetone. The shaded ribbon represents one standard deviation from the mean. Colours correspond to the different VOC classes.

### 3.3 OH reactivity, $k$(OH)

Speciated VOC concentrations alone do not indicate which individual compounds are important for $O_3$ formation. It is crucial to account for their reactivities with atmospheric oxidants, and their structure to gain insight into each individual species and compound class contribution to *in-situ* P($O_3$). All VOCs react with OH, leading to peroxy radical formation. These peroxy radical species ($HO_2$ and $RO_2$) mediate the conversion of NO to $NO_2$, leading to P($O_3$) (R1-R4). The rate at which VOCs react with OH is thus the rate determining step in the amount of $O_3$ formed.


The chemical box-model described in section 2.5 was used to investigate the total OH reactivity, expressed as $k$(OH) - a first order loss rate in units of $s^{-1}$, of observed precursors to $O_3$ (base reference model). $NO_x$, CO and individual VOC class contributions to $k$(OH), are presented in Figure 7, along with the $k$(OH) of unmeasured species generated by the model (referred to as model generated species from now on). VOCs and model generated species represented 67.4 % of the total $k$(OH), with

alkenes (9.6 %) and aromatics (8.8 %) being the largest VOC class contributors. The $k$(OH) value was higher during the night, showing an inverse relationship with boundary layer height. This is typical for urban environments where night-time emissions are typically released into a shallow boundary layer (discussed further in Stewart et al., 2021a).

Night-time boundary layer heights in Delhi were very low, leading to a clearly defined $k$(OH) profile with a maximum of ~

250 $s^{-1}$ at around 21:00, and a minimum of ~ 57 $s^{-1}$ at 14:00. The campaign average boundary layer height range was approximately 39-1550 m. Generally, $k$(OH) in megacities is found to peak at around 06:00, consistent with morning emissions into a shallow boundary layer. A small peak is also observed around this time in Delhi, but the largest peak is calculated at around 21:00. $NO_x$ represents ~ 35-40 % of the total $k$(OH) at these times, perhaps due to high volumes of traffic at these times. This is supported by VOC traffic PMF factors described in Cash et al., 2021, which peaks during the evening (~19:00-

21:00), and account for ~ 87 % of the total emissions at this time.

The values of $k$(OH) determined in this work are significantly higher than those observed in other megacities, with a daytime minimum $k$(OH) more than double that observed in Beijing during the summer of 2017 (Whalley et al., 2020). Previous studies in New York City, Mexico City and Tokyo have observed $k$(OH) >100 $s^{-1}$ (Ren et al., 2006; Shirley et al., 2005; Yoshino et

al., 2006). A maximum summertime $k$(OH) of 116 $s^{-1}$ was observed in London during rush hour, but lower OH reactivities of 22-37 $s^{-1}$ were more typical during the campaign (Whalley et al., 2016). In a study during summertime in Seoul, average $k$(OH) was *ca.* 15 $s^{-1}$ in the afternoon, increasing to 20 $s^{-1}$ at night (Kim et al., 2016). High OH reactivities were observed in the Pearl River Delta, with mean maximum reactivities of 50 $s^{-1}$ at daybreak, and mean minimums of 20 $s^{-1}$ observed at noon (Lou et al., 2010). Average summertime $k$(OH) observations from Beijing, China, observed a $k$(OH) maximum of ~ 37 $s^{-1}$ at around

06:00; and daily minimum of $k$(OH) ~ 22 $s^{-1}$ at 15:00 (Whalley et al., 2020), within a boundary layer range of approximately 150-1500 m.

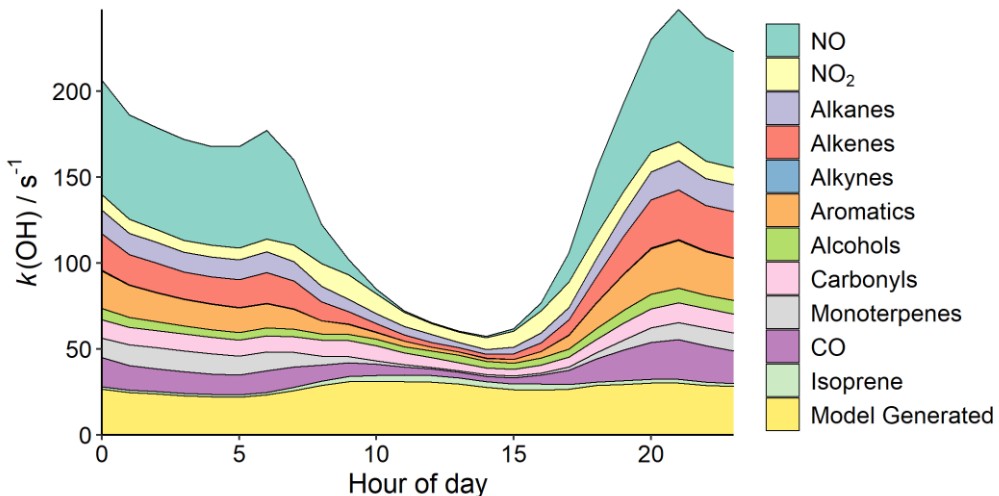

**Figure 7:** Diel profile of campaign averaged VOCs, CO and NO$_x$ contributions to OH reactivity, $k$(OH).

In Delhi, O$_3$ concentrations rapidly increase from ~ 08:00, peaking around ~ 13:00 before declining in the afternoon (Figure 3). A breakdown of the percentage contribution of each species class to morning $k$(OH) (08:00 – 12:00) is presented in Figure 8. The average morning $k$(OH) is dominated by model generated species (31.9 %), followed by NO$_x$ (21.5 %) and carbonyls (9.2 %).

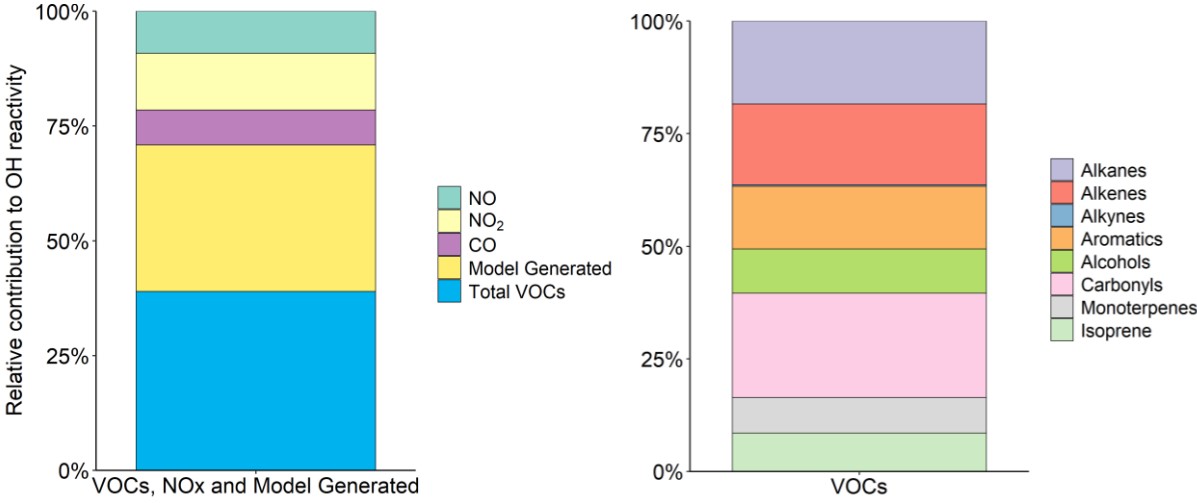


**Figure 8:** Relative contribution of species classes to average morning OH reactivity, $k$(OH), (08:00-12:00). The contribution of each group to the total $k$(OH) including NO$_x$ and model generated species (left) and the contribution of each class to just the VOC proportion of $k$(OH) (right).

## 3.4. O₃ production potentials

Model generated species are the products of reactions of VOCs of all classes, meaning the overall contribution of individual classes of VOCs to $k$(OH) is underestimated in Figures 7 and 8 (section 3.3). To assess the true contribution of VOCs to $k$(OH) and production, several model runs where each constrained VOC class was increased by 5 % were compared to the base reference model (scenario 4, sections 2.5).

One way to assess the contributions of different VOCs to O₃ production is to determine the change in O₃ production, $\Delta P(O_3)$, when the base reference model is compared to an adapted model where the VOC of interest is increased by 5 % (Elshorbany et al., 2009). The additional P(O₃) resulting from the 5 % perturbed system was averaged over 24 hours. For both the base and 5 % scenarios, the net rate of O₃ production, P(O₃), is calculated by subtracting the instantaneous rate of O₃ loss, $L(O_3)_{inst}$, by the instantaneous rate of O₃ production, $P(O_3)_{inst}$, via equations 4-6. $P(O_3)_{inst}$ was calculated by determining the NO₂ production rate through reactions of NO with HO₂ and RO₂ (Whalley et al., 2018), assuming the production of a molecule of NO₂ equates to the production of a molecule of O₃ via reaction R5. NO₂ loss processes which do not yield O₃, such as removal by reaction with OH and reaction with acyl peroxy radicals to form peroxy acetyl nitrates (PANs), must also be accounted for. In these calculations, OH, HO₂ and RO₂ concentrations are generated from the observationally constrained model.

$P(O_3)_{inst}$ is the rate of NO oxidation by HO₂ and RO₂ radicals:

$$P(O_3)_{inst} = \left( k_{HO_2+NO}[HO_2][NO] + \sum_i k_{RO_{2i}+NO}[RO_2][NO] \right) \tag{4}$$

$L(O_3)_{inst}$ is the rate of loss of O₃ though reactions with OH, HO₂, and photolysis followed by a reaction with H₂O vapour, along with the loss of NO₂ through reactions with OH:

$$L(O_3)_{inst} = j(O^1D)[O_3] * f + k_{OH+O_3}[OH][O_3] + k_{HO_2+O_3}[O_3][HO_2] + k_{OH+NO_2+M}[NO_2][OH][M] \tag{5}$$
$$+ \sum_i k_{RO_{2i}+NO_2+M}[RO_2][NO_2][M]$$

where $f$ is the fraction of O($^1$D) atoms (formed in the photolysis of O₃) that react with H₂O vapour to form OH, rather than undergo collisional stabilisation. Net O₃ production, $P(O_3)$ can then be calculated as:

$$P(O_3) = P(O_3)_{inst} - L(O_3)_{inst} \tag{6}$$

The O$_3$ production resulting from an incremental increase of each VOC between 08:00 and 12:00, $P(O_3)_{increm}$, was then calculated with equation 7:

$$\Delta P(O_3)_{increm} = Mean\ (P(O_3)_i - P(O_3)_{base}) \tag{7}$$

where $P(O_3)_i$ is the mean O$_3$ production between 08:00 and 12:00, calculated from the model run where the VOC of interest is increased by 5 %. Using this approach, the ten VOCs contributing to the greatest change in P(O$_3$) on an incremental increase, $P(O_3)_{increm}$, are isoprene, α-terpinene, ethene, toluene, propene, α-phellandrene, ethanol, 1,2,4-trimethylbenzene, CO and β-ocimene (table 1).

**Table 1**: Top 10 highest ΔP(O$_3$)$_{increm}$ VOCs, and their respective classes, averaged between 08:00 and 12:00.

| Species | Class | ΔP(O$_3$)$_{increm}$ / ppb h$^{-1}$ | P(O$_3$) increase / % |
|---|---|---|---|
| Isoprene | Isoprene | 0.94 | 0.74 |
| α-Terpinene | Monoterpene | 0.66 | 0.53 |
| Ethene | Alkene | 0.40 | 0.32 |
| Toluene | Aromatic | 0.37 | 0.30 |
| Propene | Alkene | 0.35 | 0.27 |
| α-Phellandrene | Monoterpene | 0.30 | 0.24 |
| Ethanol | Alcohol | 0.30 | 0.24 |
| 1,2,4-Trimethylbenzene | Aromatic | 0.29 | 0.23 |
| CO | – | 0.28 | 0.22 |
| β-Ocimene | Monoterpene | 0.26 | 0.21 |

### 3.5 P(O$_3$) sensitivities to VOC/NO$_x$ ratios

To better understand the complex, non-linear chemistry at play, a box model is used to probe the chemical sensitivities of observed precursors to O$_3$ formation. The model was run 144 times, each with adjusted concentrations of VOCs and NO$_x$ (scenario 1). The observed concentrations were multiplied by a factor to generate unique model runs from which $P(O_3)$ was calculated using equations 4-6. The resultant P(O$_3$) isopleth is presented in figure 9.

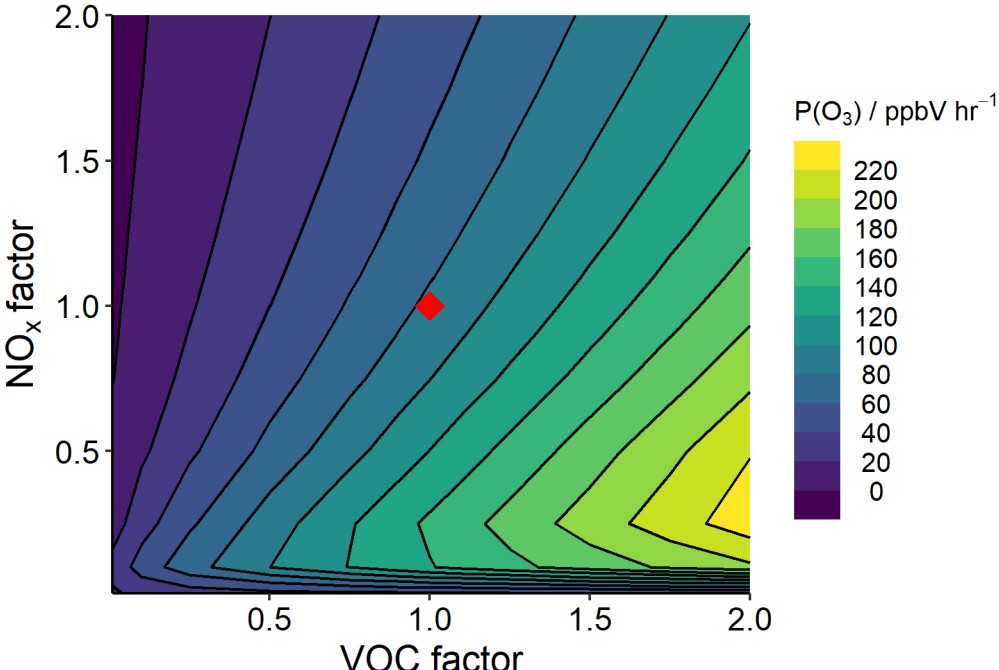


**Figure 9:** Mean modelled $O_3$ production, $P(O_3)$, isopleth between 08:00 and 12:00 based on varying VOC and $NO_x$ concentrations in the model. The red diamond at point 1,1 represents modelled $P(O_3)$ at observed VOC and $NO_x$ concentrations.

The modelled VOC-$NO_x$ $P(O_3)$ isopleth supports the assignment of Delhi being, on average, in a VOC-sensitive photochemical

regime (Sillman et al., 1990), with the diel profile of $O_3$ production peaking at 09:00 (see supplementary, figure S3). Reducing $NO_x$ alone and maintaining VOC concentrations would result in an increase in $P(O_3)$. Therefore reducing, or even maintaining, $O_3$ levels in the future will require a reduction in VOC emissions, if future emission control measures continue to target $NO_x$ emissions in Delhi. This is consistent with a study of observational data in Delhi, whereby a SARS-CoV-2 lockdown led to a reduction in $NO_x$ emissions and concentrations and an increase in the concentration of $O_3$ (Jain and Sharma, 2020). To

implement an efficient and realistic VOC reduction plan, the key VOCs contributing to $P(O_3)$ need to be identified, along with their sources.

As has been previously discussed, $O_3$ concentrations limits of 50 ppbV were regularly exceeding during the campaign, with the maximum daily 8-hour averages peaking at 88 ppbV (Figure 4). To successfully reduce $O_3$ to the limit of 50 ppbV, $O_3$

production must be reduced by 56%. This can be achieved by reducing $NO_x$ by 25%, 50% and 75% along with a concurrent reduction in VOCs of 48%, 61% and 78% respectively. Alternatively, if VOCs were halved, $NO_x$ reductions could not exceed 29% to peak $O_3$ below 50 ppbV. To obtain a reduction in $O_3$ production without reducing VOCs would require a $NO_x$ reduction of at least 92%. However, it is important to consider that reducing emissions of $NO_x$ and VOCs by these values will only

impact the in situ formation of $O_3$. Regional $O_3$ production leading to the transportation of $O_3$ from outside of Delhi is beyond
the scope of this analysis and must also be taken into consideration by policy makers.

### 3.6 P($O_3$) sensitivity to VOCs: by class

The impact of changing VOC concentrations (by class) on mean P($O_3$) was investigated using scenario 3 (section 2.5). For each model run, the constrained concentrations of all species in the class of interest were multiplied by a "VOC factor". Modelled P($O_3$) was calculated upon changing "VOC factor" for CO and six different VOC classes: alkanes, alkenes, aromatics, monoterpenes, isoprene and alcohols. Changes in P($O_3$) were found to be most sensitive to changes in aromatics, followed by the monoterpenes and alkenes (Figure 10). Halving aromatic, monoterpene and alkene concentrations independently (reducing VOC factor = 1, to VOC factor = 0.5, Figure 10) reduced modelled P($O_3$) by 15.6 %, 13.1 % and 12.9 % respectively. However, future air quality control strategies are also likely to include a co-reduction in $NO_x$. As we observed in Figure 9, a reduction in $NO_x$ coupled with an insufficient reduction in VOCs is likely to increase P($O_3$) in Delhi, under VOC-limited conditions. On concurrently reducing VOC class and $NO_x$ by half (factor = 0.5, figure 10), aromatics, alkenes and monoterpenes lead to the smallest increase in P($O_3$) (24.9 %, 29.8 % and 35.5 % respectively). However, this still represents a significant increase in P($O_3$). This suggests targeting one VOC class alone in future pollutant reduction strategies is insufficient, and that reducing a source/sources emitting multiple VOC classes is important to avoid an increase in P($O_3$) on simultaneously reducing $NO_x$.

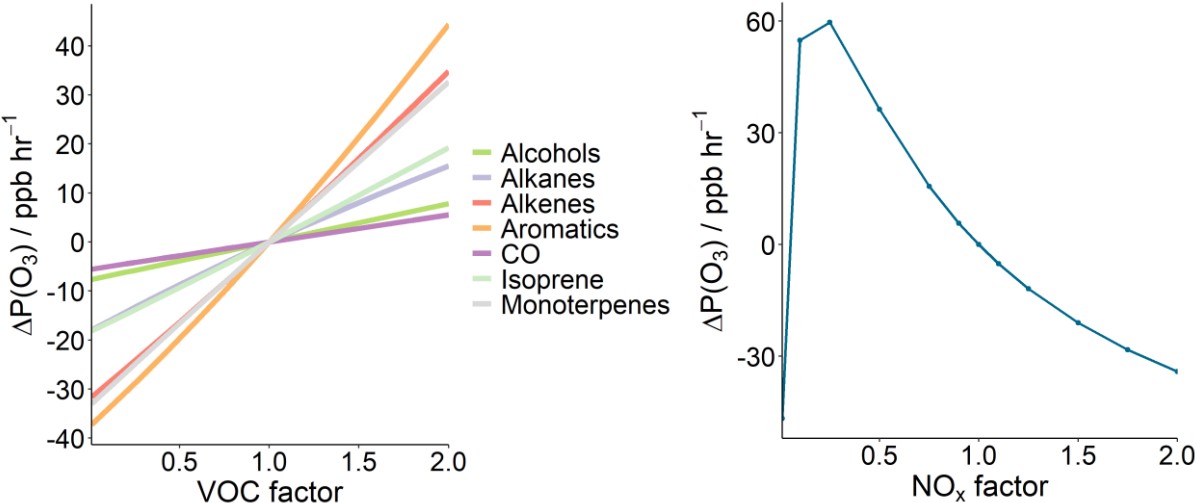

Figure 10: Modelled mean morning (08:00 – 12:00) change in P($O_3$) rate for CO and six VOC classes upon varying their concentrations by multiplying by factor, "VOC factor" (left) and by vary just $NO_x$ (right).

### 3.7 The impact of aerosol surface area on P(O₃)

Aerosol particles in the troposphere can impact gas phase chemistry and photochemical activity, affecting P(O₃), in a number of ways. Aerosols can interact with incoming solar radiation, either scattering or absorbing sunlight. The precise impact of aerosol is dependent on a range of factors including chemical composition, particle size distribution and phase state. It has previously been shown that in highly polluted urban areas, attenuation of the actinic flux due to aerosol absorption can significantly reduce photolysis rates, and hence P(O₃), by up to 30 % (Castro et al., 2001; Hollaway et al., 2019; Real and

Sartelet, 2011; Wang et al., 2019). Conversely, aerosol scattering can potentially increase P(O₃) by increasing the photolysis rates (Dickerson et al., 1997; He and Carmichael, 1999).

    Aerosol can also participate in heterogeneous chemistry, i.e., uptake of radicals to the aerosol or reactions at the aerosol surface can affect gas phase radical budgets (George et al., 2015). A recent regional modelling study (Li et al., 2019a) linked increasing

O₃ concentration trends between 2013 and 2017 in Beijing to decreasing PM$_{2.5}$ concentrations. The study attributed this to the decreased uptake of HO₂ radicals to aerosol particles, leading to increased HO₂ available to participate in P(O₃) (see R4). However, an experimental study (Tan et al., 2020) in the North China Plain did not observe the effect. The relationship between aerosol optical depth (AOD) and photolysis rates is strongly non-linear and has a larger impact at low solar zenith angles (i.e., in the morning/evening /high latitudes) (Wang et al., 2019).


    To investigate the potential impact of aerosol related processes in Delhi, aerosol surface area (ASA) and photolysis rates were varied independently in the model using a scaling factor ("Factor", figure 11), and the resulting impact on P(O₃) calculated. The first order loss of HO₂ to aerosol surface area ($k$) was calculated using equation 3 (scenario 5, section 2.5).

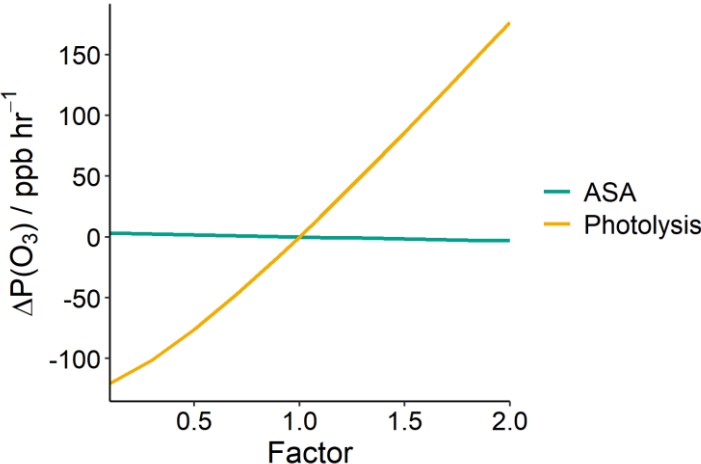


**Figure 11:** Impact of varying photolysis rates and aerosol surface area (ASA) on mean modelled morning O₃ production rates, ΔP(O₃) (08:00 – 12:00).

Changes in ASA, with respect to $HO_2$ uptake, were found to have minimal impact on the modelled mean morning $P(O_3)$ rate (08:00 – 12:00), as shown in Figure 11, in line with the observations of Tan et al., 2020. The lifetime of $HO_2$ is dominated by its reaction with NO, within the campaign averaged NO concentrations observed. Changes to the photolysis rates have a large impact on $P(O_3)$ (Figure 11), with $P(O_3)$ increasing roughly linearly with increasing photolysis rates. The impact of changes to AOD on photolysis rates are non-linear and dependent on solar zenith angle, SZA, (i.e., time of day). This work shows that the majority of daily $P(O_3)$ occurs between 08:00 and 12:00, representing an SZA range of 35 - 80°. Aerosol loading is expected to reduce in Delhi going into the future if India implements air pollution controls in line with other countries. Based on the work of Wang et al. (2019), reductions in PM, leading to reductions in AOD, could potentially increase $j(NO_2)$ by up to 30 % at mid-day and up to 100 % at 08:00. This will lead to increased photochemical activity and higher $P(O_3)$.

From an air quality perspective, it is important to consider reducing not only VOCs, but also $NO_x$ concentrations in Delhi. Along with this, reductions in PM should also be considered. As we have seen in Figure 11, reducing ASA does not significantly impact $P(O_3)$ through heterogenous chemistry. However, an ASA reduction will result in a decrease in AOD, which will in turn increase photolysis rates. A recent study by Chen et al., 2021 modelled the impact of AOD on photolysis in Delhi during November 2018, and found that halving the AOD results in a 14 % increase in $j(NO_2)$. In Delhi, total VOCs would need to be reduced by ~ 50 % to maintain current $P(O_3)$ rates if $NO_x$ were halved and photolysis rates increased by 14 %. However, the impact of reducing aerosol on photolysis rates is complex. Reductions in aerosol at ground level may lead to either increased or decreased photolysis rates near the surface, depending on the scattering properties of the aerosol. Changes in photolysis rates from increased or decreased aerosol loading may also vary throughout the depth of the boundary layer. The box model assumes photolysis rates are uniform throughout the boundary layer and that aerosols are well mixed. A more detailed study into the temporal and spatial patterns of aerosol in Delhi and its impact on photolysis rates is required to fully assess the aerosol impact on in situ $O_3$ production (Castro et al., 2001).

It is important to note that this study focuses on the sensitivity of $P(O_3)$ to ASA through $HO_2$ uptake only. Additional chemical consequences and feedbacks of decreasing aerosol, such as changes to HONO concentrations, have not been accounted for here. With the high levels of $NO_2$ observed in Delhi, the heterogenous conversion of $NO_2$ to HONO on particle surfaces may be an important mechanism (Liu et al., 2014; Lee et al., 2016; Tong et al., 2016; Lu et al., 2018b). HONO reductions from decreased ASA may lead to reduced OH radical formation in Delhi, thus the impact of ASA reduction on $P(O_3)$ may be underestimated in this work.

**3.8 Discussion**

If future air pollution controls in Delhi follow the air quality strategies prevalent or planned in many other countries, including the EU, there is a danger that urban $O_3$ concentrations could significantly increase, unless careful consideration is given to the specific atmospheric chemistry occurring in the city. Many other major cities across the globe have focused their air pollution abatement strategies on reducing $NO_x$ and particulate emissions from traffic sources. Urban $NO_x$ emissions (at tailpipe) are likely to decline over time, as a result of improved exhaust gas treatments, the turnover of the fleet to newer, less polluting

vehicles, and the increasing prevalence of electric vehicles (Molina, 2021). While this is also likely to reduce ambient VOC concentrations through reduction in tailpipe and evaporative emissions, this reduction may be smaller than for $NO_x$. The magnitude of resulting changes in $P(O_3)$ from reduced road transport emissions may depend on the proportion of VOCs emitted from road transport relative to other, non-vehicular sources. As demonstrated here, reductions in $NO_x$ without sufficiently reducing VOC emissions may lead to large increases in $P(O_3)$ under a continued VOC-limited regime. Reducing traffic

emissions will also likely lead to reduced aerosol loading in Delhi. Our study suggests a reduction in aerosol surface area will have very little direct effect on $P(O_3)$ via heterogenous chemistry, as $HO_2$ reactivity is dominated by its reaction with NO. However, an ASA reduction is likely to increase the amount of sunlight reaching the boundary layer, and hence photolysis rates. This will lead to a subsequent increased $P(O_3)$, as discussed in more detail by Chen et al., 2021.

$P(O_3)$ in Delhi was found to be most sensitive to reductions in aromatics and alkenes, and so monitoring the abundance and knowing the sources of these compounds in Delhi is essential for implementing effective pollutant reduction strategies to avoid a future rise in $P(O_3)$. As these classes are thought to come mainly from traffic sources, it is possible that reducing road transport emissions may reduce traffic-derived VOCs sufficiently, along with reducing $NO_x$, to perturb a large increase in $P(O_3)$. However, the proportion of VOCs in Delhi, and at IGDTUW, attributed to traffic sources is uncertain, and thus the extent to

which road transport reductions will impact $P(O_3)$ is unknown.

More information of the effect of reductions in $NO_x$ and VOCs by source can be obtained by varying sources described by emissions inventories. Anthropogenic emission of $NO_x$, CO and VOCs by source in Delhi are available from the EDGAR v5.0 Global Air Pollutant Emissions and EDGAR v4.3.2_VOC_spec inventories (https://edgar.jrc.ec.europa.eu/#). According to the inventory, all pollutants investigated in this analysis can be almost entirely described by 5 source sectors (table 2): Road

Transport (RT); Railways, Pipelines and Off-Road Transport (RPORT), Energy for Buildings (EB), Combustion for Manufacturing (CM) and Process Emissions (PE). However, it should be noted here that the EDGAR emissions inventory describes a coarse, low spatial resolution, city-wide representation of emissions from Delhi. For this analysis, we assume the EDGAR sector split ratios for Delhi are representative of those at the IGDTUW measurement site.

Model constrained concentrations were varied by their contributions to sources in the EDGAR emissions inventory (scenario 7, section 2.5). Data from the inventory represents average annual emissions from source sectors, last updated for VOCs in

2012 and 2008 for monoterpenes, with 0.1° x 0.1° spatial resolution. VOCs were assumed to be described in full by all sources available from the inventory, with no biogenic influence. Isoprene was excluded from the analysis as it is assumed to have an entirely biogenic source. As a significant anthropogenic signature is seen from the monoterpenes, and because of their

significance to $P(O_3)$ in this study, 50 % are assumed to be from anthropogenic sources found in the EDGAR emission inventory for the purpose of this analysis. A full list and description of sources used from the EDGAR inventory can be found in the supplementary. Figure 12 shows the change in $P(O_3)$ rate when reducing VOCs and $NO_x$ contributions from individual source sectors.

**Table 2**: Relative proportion of pollutants emitted from five EDGAR emission inventory source sectors: Road Transport (RT); Railways, Pipelines and Off-Road Transport (RPORT), Energy for Buildings (EB), Combustion for Manufacturing (CM) and Process Emissions (PE).

| Species Class | RT / % | RPORT / % | EB / % | CM / % | PE / % | Sum / % |
|---|---|---|---|---|---|---|
| CO | 85.0 | 2.5 | 8.1 | 4.1 | 0.1 | **99.9** |
| $NO_x$ | 60.9 | 28.9 | 0.6 | 2.5 | 0.0 | **92.9** |
| Alkanols | 0.0 | 0.0 | 0.4 | 0.2 | 99.3 | **100.0** |
| Benzene | 10.9 | 0.0 | 47.4 | 34.3 | 0.3 | **92.9** |
| Butanes | 2.2 | 0.0 | 0.3 | 4.9 | 92.5 | **99.9** |
| Dimethylbenzenes | 12.3 | 0.0 | 10.5 | 18.8 | 57.8 | **99.5** |
| Ethane | 2.4 | 0.0 | 59.7 | 32.1 | 0.2 | **94.5** |
| Ethene | 5.2 | 0.0 | 57.5 | 31.8 | 1.0 | **95.5** |
| Ethyne | 0.8 | 0.0 | 58.8 | 33.1 | 3.9 | **96.5** |
| Hexanes and higher alkanes | 30.2 | 0.0 | 8.4 | 8.6 | 52.2 | **99.3** |
| Methylbenzene | 10.2 | 0.0 | 29.0 | 31.4 | 25.8 | **96.4** |
| Monoterpenes* | 0.0 | 0.0 | 0.0 | 1.9 | 98.1 | **100.0** |
| Other alkenes and alkynes | 18.5 | 0.0 | 46.2 | 28.1 | 2.4 | **95.3** |
| Other aromatics | 49.1 | 0.1 | 19.0 | 16.6 | 10.8 | **95.6** |
| Pentanes | 42.6 | 0.0 | 0.1 | 50.0 | 6.6 | **99.3** |
| Propane | 1.3 | 0.0 | 18.0 | 29.5 | 49.7 | **98.4** |
| Propene | 15.7 | 0.0 | 48.6 | 30.2 | 1.0 | **95.5** |
| Trimethylbenzenes | 67.4 | 0.0 | 0.5 | 0.0 | 31.1 | **98.9** |

*The percentage of monoterpenes attributed to these sources in varied in the model, and is assumed to be 50 % in the base case.

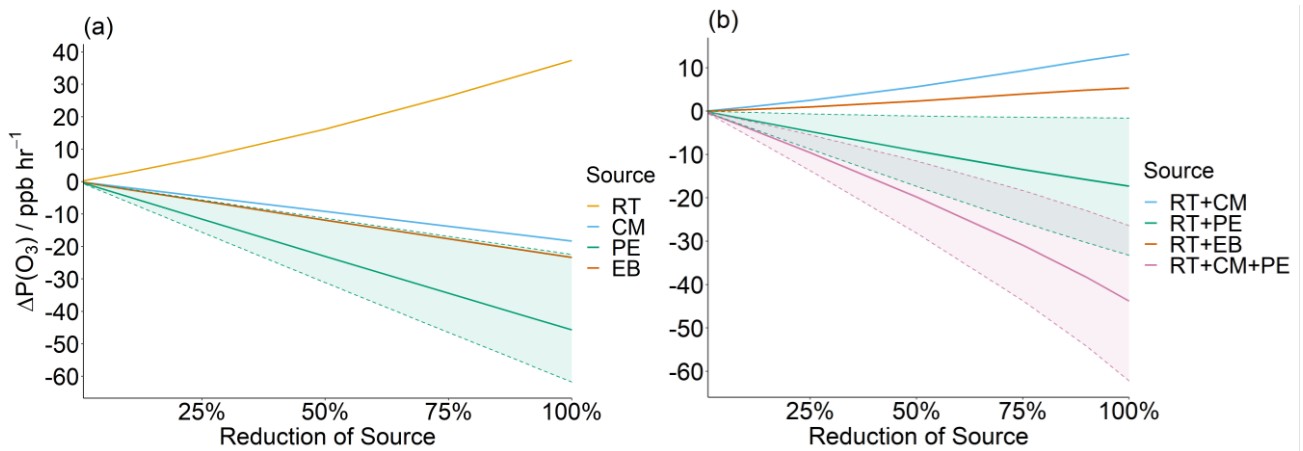

**Figure 12:** Change in $P(O_3)$ on reducing road transport (RT), combustion for manufacturing (CM), process emissions (PE) and energy for buildings (EB) (left); and (right) reducing road transport simultaneously along with either i) CM, ii) PE, iii) EB, iii) CM and PE. The shaded region, bounded by a dashed line, represents the variability in $\Delta P(O_3)$ when 0-100 % of observed monoterpenes contribute to anthropogenic EDGAR source sectors that include PE. The solid lines represent the base case whereby 50 % of monoterpenes are assumed to be from anthropogenic sources described in the EDGAR inventory (the other 50 % is assumed to be from biogenic sources).

Based on this analysis, reducing the RT source in isolation results in increased $P(O_3)$, whilst reducing CM, PE and EB independently leads to decreased $P(O_3)$. This is explained by CM, PE and EB being major sources for VOCs, and RT also being a major source of $NO_x$. Although reducing RT also reduces some VOCs, particularly aromatic and higher alkane species, there is not sufficient VOC reduction to compensate for the large co-reduction in $NO_x$, leading to increased $P(O_3)$ (Figure 10). Reducing RT by 100 % would still lead to a modelled $P(O_3)$ increase of ~ 40 ppb $h^{-1}$ (Figure 12a). It is important to acknowledge the uncertainties in this analysis. The EDGAR emissions inventory attributes a low percentage split of VOCs to the RT source sector at the city level, perhaps underestimating the proportion of VOCs that would be reduced with RT reductions at the measurement site.

Air quality mitigation strategies should therefore focus on reducing RT, alongside one or more additional major VOC sources. For example, Figure 12b shows $P(O_3)$ begins to reduce when RT emissions are reduced along with PE. These reductions are even greater when RT is reduced with both PE and CM sources. When RT is reduced by 50 % alongside PE and PE+CM, modelled $P(O_3)$ is reduced by ~ 10 ppb $h^{-1}$ and ~ 20 ppb $h^{-1}$ respectively. The VOCs that contribute the most to PE and CM emissions are *n*-butane, propane, alcohols, toluene and xylenes. Although alcohols and alkanes were not identified as the key VOC classes contributing to $P(O_3)$ (section 3.6), they are the two largest classes of VOCs observed in Delhi by mass, making up 42 % and 18 % of the total measured VOCs, respectively (Figure 5). Reducing the PE source thus leads to large reductions

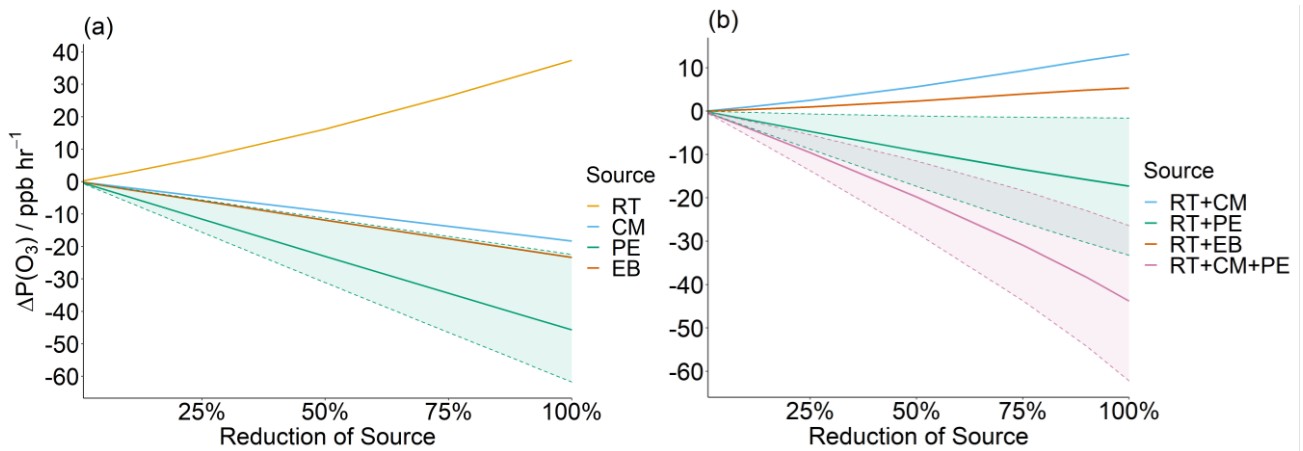

in VOCs by mass, helping to drive down P(O₃). However, it is worth noting that the effectiveness of reducing the RT + PE source on modelled P(O$_3$) is dependent on the proportion of anthropogenic monoterpene emissions in Delhi. According to the EDGAR emission inventory, 98.1% of anthropogenic monoterpenes in Delhi are attributed to process emissions (PE) (Table 2). These emissions include sources such as emissions from chemical industry, and other industrial processes, and include solvent emissions and emissions from product use (see Table S3 in supplementary). As emissions from these sources are grouped together in EDGAR inventory, the exact sources from which monoterpenes are attributed to cannot be identified. The sensitivity of ΔP(O$_3$) from reducing process emission sources (PE, RT+PE, and RT+PE+CM) is shown by the shaded regions in Figure 12, where the dashed lines represent the sensitivity limits where the observed monoterpenes are between 0% and 100% anthropogenic (as opposed to biogenic). There is relatively little impact on P(O$_3$) on reducing RT+PE when monoterpenes are assumed to have an entirely biogenic source. However, it is clear that although the degree to which reducing process emissions along with road transport in this study impacts P(O$_3$) cannot be accurately determined, even if no monoterpenes are reduced within this source, reducing it does not negatively impact P(O$_3$). It is also important to consider possible under-estimations for the monoterpene contribution to RT. Monoterpene observations in Delhi were strongly correlated with CO emissions, suggesting anthropogenic sources (Stewart et al., 2021a). The EDGAR emissions inventory assigns 0% of the anthropogenic monoterpenes in the inventory to the RT source sector (table 2). An analysis of the PTR-QiTOF flux data, obtained at the IGDTUW measurement site directly after the concentration measurement period ended, suggests ~ 60% of the monoterpenes observed could be attributed to traffic factors (Cash et al., 2021). A study by Wang et al, 2020 suggested vehicular and burning sources may contribute to the anthropogenic emissions of biogenic molecules. Other possible sources of anthropogenic monoterpenes in Delhi are emissions from cooking herbs and spices, and from fragrances and personal care products (Klein et al., 2016, McDonald et al., 2018).

It should be noted that the EDGAR inventory data used in this study is cropped to the Delhi area, and thus some anthropogenic sources may be missing. One example of this is that of agricultural burning, which is frequent in areas surrounding Delhi and across the Indo Gangean Plain (IGP) (Jat and Gurjar, 2021; Kulkarni et al., 2020). It is likely that regional sources of O₃ and its precursors will have a significant effect on Delhi's air quality, and vice versa. There are likely to be many missing sources from the EDGAR emissions inventory, along with discrepancies between the top-down and bottom-up data. In addition, we assume the ratios of VOCs and NO$_x$ in each source sector for the Delhi region are representative of the ratios emitted near IGDTUW. Whilst the OH reactivity analysis provides an instantaneous assessment of a VOC's reactivity, based on local observations with a small spatial footprint, the EDGAR Delhi-wide sectoral split represents an aggregate of a wider region and may under-represent the RT VOC contribution observed at the measurement site. As a result, the magnitude of the modelled increased P(O$_3$) when road transport is reduced may represent a worst-case scenario.

**4 Conclusions**

A detailed chemical box model constrained to an extensive observational dataset of 86 VOCs, 34 photolysis rates, NO, $NO_2$, CO, $SO_2$, HONO, temperature, pressure and relative humidity was used to explore the sensitivity of photochemical $O_3$ production, $P(O_3)$, to VOCs and $NO_x$ in the Indian megacity of Delhi. The urban measurement site at the Indira Gandhi Delhi Technical University for Women is determined to be in a VOC-limited chemical regime. Our analysis examined the sensitivity of VOC classes to mean morning $P(O_3)$, and the aromatic VOC class was identified as being the most important, with a 50 % reduction in ambient concentrations leading to a reduction in modelled morning $P(O_3)$ of 15.6 %, followed by monoterpenes

and alkenes (13.1 % and 12.9 % respectively). The direct impact of the aerosol burden on $P(O_3)$ was found to be negligible with regards to heterogeneous radical uptake. However, $P(O_3)$ was sensitive to increasing photolysis, which may result from decreasing particulate matter that is likely to arise from future emission reduction strategies. Though it is important to reduce $NO_x$ and particulate matter in all abatement strategies, reducing $NO_x$ without reducing VOCs was found to significantly increase $P(O_3)$. VOCs and $NO_x$ were also evaluated by their respective contributions to different source sectors as defined in

the EDGAR emissions inventory. Reducing emissions from road transport sources alone lead to increased $P(O_3)$, even when the source was removed in its entirety. This suggests a balanced approach to pollution reduction strategies is required, and multiple sources should be targeted simultaneously for effective reduction, incorporating major monoaromatic, alkene and anthropogenic monoterpene VOC sources. Reducing road transport emissions along with combustion from manufacturing and process emissions was found to be an effective way of reducing $P(O_3)$. Monoterpenes are found to have a significant impact

on $P(O_3)$ and showed a diurnal profile consistent with other anthropogenic VOCs. Future work should be carried out to determine the sources and fraction of anthropogenic emissions to the observed monoterpene concentrations measured in Delhi, with a recent study suggesting 60% of the monoterpenes observed at the site are coming from traffic related sources (Cash et al., 2021, in preparation). To further understand the complex urban atmospheric chemistry in Delhi, more model constraints are required, such as the measurement of radical species and $k$(OH). The potential effects of chlorine chemistry were also not

investigated in this study, and may have an important role in the local chemistry due to the prevalence of waste burning in the city (Gunthe et al., 2021). Satellite observations and aircraft measurements may also help develop our understanding of the regional impacts of regional agricultural biomass burning on Delhi from neighbouring states. This work highlights that a careful approach, considering the complexities of chemical processing in the urban atmosphere, is required for effective air quality improvement strategies.

*Data availability.* Data used in this study can be accessed from the CEDA archive: Nemitz, E., Acton, W. J., Alam, M. S., Drysdale, W. S., Dunmore, R. E., Hamilton, J. F., Hopkins, J. R., Langford, B., Nelson, B. S., Stewart, G. S., Vaughan, A. R., Whalley, L. K., (APHH India) Megacity Delhi atmospheric emission quantification, assessment and impacts (DelhiFlux), https://catalogue.ceda.ac.uk/uuid/ba27c1c6a03b450e9269f668566658ec, 2020.

*Author contribution.* BSN prepared the manuscript, with contributions from all authors. BSN, GJS, WSD, ARV, RED, WJA, LRC, MSA, BL, EN and JRH provided measurements and data processing of the comprehensive suite of atmospheric pollutants used in this study. MJN, PME, ES, LKW, DEH, RS and SC provided modelling assistance and model code development used in the chemical box model. UAS, DCSB, LKW, MSA and WJB contributed to the data processing of measured chemical species. JMC contributed to scientific discussion, critical to the findings of this work. S, RG, BRG and EN assisted with the logistics of lab set-up at the measurement site, and data analysis. ACL, JFH, CNH, WJB, JRH, ARR and JDL provided overall guidance to experimental setup and design, along with assistance in operating instrumentation, and data analysis and interpretation.

*Competing interests*. The authors declare that they have no conflict of interest.

*Disclaimer.* The paper does not discuss policy issues, and the conclusions drawn in the paper are based on interpretation of results by the authors and in no way reflect the viewpoint of the funding agencies or institutions authors are affiliated with.

*Acknowledgements.* The authors acknowledge Dr. Pawel Misztal and Dr. Brian Davison for their involvement in operating the PTR-TOF, and Dr. Tuhin Mandal at CSIR-National Physical Laboratory for his support in facilitating the measurement sites used in this project. This work was supported by the Newton Bhabha fund administered by the UK Natural Environment Research Council, through the DelhiFlux and ASAP projects of the Atmospheric Pollution and Human Health in an Indian Megacity (APHH-India) programme. The authors gratefully acknowledge the financial support provided by the UK Natural Environment Research Council and the Earth System Science Organization, Ministry of Earth Sciences, Government of India, under the Indo-UK Joint Collaboration (grant nos. NE/P016502/1, NE/P016499/1 and MoES/16/19/2017/APHH) (DelhiFlux). Beth S. Nelson and Gareth J. Stewart acknowledge the NERC SPHERES doctoral training programme for studentships. James M. Cash is supported by a NERC E3 DTP studentship.

*Financial support*. This research has been supported by the Natural Environment Research Council (grant no. NE/P016502/1 and NE/P016499/1) and the Ministry of Earth Sciences (grant no. MoES/16/19/2017/APHH, DelhiFlux)

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
