# Peer review of "In situ ozone production is highly sensitive to volatile organic compounds in Delhi, India"

_Atmospheric Chemistry and Physics, 2021_

## Author Comment (AC1)

The authors would like to thank the reviewers for their positive and constructive comments on this manuscript. Reviewer comments are in **bold**, followed by the author response and resulting text amendments.

**Reviewer 1: Major comments**

**1) The background material in the Introduction is largely out of date and here I list some references that provide a current assessment of ozone's distribution and trends. When discussing the impacts of ozone on human health, Jerrett et al. is a good reference, but it's quite old. Fleming et al. 2018 from the Tropospheric Ozone Assessment Report (TOAR) provide an overview of ozone's health impacts. In terms of ozone increases since the mid- 20th century, the earlier findings of Parrish et al. 2014 have now been superseded by Tarasick et al. (2019) (from TOAR). Since the 1990s, surface ozone trends vary by region (Gaudel et al., 2018; Cooper et al., 2020; Lu et al., 2020), but in the free troposphere trends since the 1990s have been overwhelmingly positive (Gaudel et al., 2020; Liao et al., 2020; also see the review by Cooper et al., 2020). The paper by Ni et al. (2018) is not a good reference regarding ozone trends as it only focuses on a single year (2008). A good paper that shows the increases of ozone across China is Lu et al. 2020.**

*Response*: The reviewer makes a good point that there are more up to date references than those used in the introduction section of the manuscript. To improve this, additional text has been added to this section, along with updated references as suggested by the reviewer. The authors would like to thank the reviewer for kindly providing some excellent references to improve this section.

*Text change*: Tropospheric $O_3$ is both an air pollutant and an important greenhouse gas throughout the troposphere (Skeie et al., 2020). High levels of $O_3$ can adversely affect vegetation, global crop yields (Avnery et al., 2011) and human health, with long-term exposure increasing the risk of death from cardiovascular and respiratory illnesses (Jerrett et al., 2009), and short-term exposure leading to the exacerbation of asthma in children (Thurston et al., 1997). $O_3$ exposure has been linked to both acute and chronic pulmonary and cardiovascular health outcomes through both animal toxicological and human clinical studies, with one study showing statistically significant decreases in the lung function of adults on an average exposure of 70 ppbV of $O_3$ across five 6.6-hour windows (WHO 2005 and 2013, Schelegle et al., 2009, EPA 2013, Fleming et al., 2018).

As a result of increased anthropogenic emissions, tropospheric $O_3$ increased globally during the 20th century, and has continued to rise regionally in Asia during the 21st century (Fleming et al., 2018; Royal Society, 2008, Lu et al., 2020). Background tropospheric $O_3$ has also continued to increase (Parrish et al., 2014, Tarasick et al., 2019). Since the 1990s, surface $O_3$ trends have varied by region (Gaudel et al., 2018, Cooper et al., 2020, Lu et al., 2020), but trends in the free troposphere have been overwhelmingly positive (Gaudel et al., 2020, Liao et al., 2020). Both satellite data and global chemical transport models have identified India and East Asia as the region with the greatest $O_3$ increases between 1980-2016 (Ziemke et al., 2019), with the rate of change per decade between 2005-2016 more than double that of the rate between 1979-2005 (Ziemke et al., 2019).

**2) According to the ACP/Copernicus Data Policy, the paper needs to include a "Data availability" section, as follows: Authors are required to provide a statement on how their underlying research data can be accessed. This must be placed as the section "Data availability" at the end of the manuscript. Please see the manuscript preparation guidelines for authors for the correct sequence. If the data are not publicly accessible, a detailed explanation of why this is the case is required. The best way to provide access to data is by depositing them (as well as related metadata) in FAIR-aligned reliable public data repositories, assigning digital object identifiers, and properly citing data sets as individual contributions. The authors have not provided a "Data availability" section, which needs to be addressed before the paper can be published. Further details are available here: https://www.atmospheric-chemistry-and-physics.net/policies/data_policy.html**

*Response:* A data availability section has been added, proving a catalogue link to the project data on the CEDA archive.

*Text change*: *Data availability*. Data used in this study can be accessed from the CEDA archive: Nemitz, E., Acton, W. J., Alam, M. S., Drysdale, W. S., Dunmore, R. E., Hamilton, J. F., Hopkins, J. R., Langford, B., Nelson, B. S., Stewart, G. S., Vaughan, A. R., Whalley, L. K., (APHH India) Megacity Delhi atmospheric emission quantification, assessment and impacts (DelhiFlux), https://catalogue.ceda.ac.uk/uuid/ba27c1c6a03b450e9269f668566658ec, 2020.

**3) The authors conducted a range of sensitivity tests to understand the response of ozone production to changes in NOx and VOCs. However, an air quality manager who is tasked with keeping ozone levels below the Indian ozone standard of 50 ppbv needs more information. They need to know how much they need to cut NOx and VOCs in order to keep the maximum daily 8-hour average below 50 ppbv. To make the study more relevant to air quality management the authors should experiment with their box model to find a range of NOx and VOC mixing ratios that will keep ozone below 50 ppbv.**

*Response:* The authors recognise that more information on possible ways $NO_x$ and VOCs can be reduced to achieve ozone levels of 50 pbbV would be useful to policy makers and those interested in air quality management. A more detailed description of the isopleth has been added to section 3.5, to provide information on how much VOCs would need to be reduced by to ensure the maximum 8-hour average observed during the campaign is reduced to 50 ppbV on reducing $NO_x$ by 25%, 50% and 70%.

*Text change:* As has been previously discussed, $O_3$ concentrations limits of 50 ppbV were regularly exceeding during the campaign, with the maximum daily 8-hour averages peaking at 88 ppbV (Figure 4). To successfully reduce $O_3$ to the limit of 50 ppbV, $O_3$ production must be reduced by 56%. This can be achieved by reducing $NO_x$ by 25%, 50% and 75% along with a concurrent reduction in VOCs of 48%, 61% and 78% respectively. Alternatively, if VOCs were halved, $NO_x$ reductions could not exceed 29% to peak $O_3$ below 50 ppbV. To obtain a reduction in $O_3$ production without reducing VOCs would require a $NO_x$ reduction of at least 92%. However, it is important to consider that reducing emissions of $NO_x$ and VOCs by these values will only impact the in situ formation of $O_3$. Regional $O_3$ production

leading to the transportation of $O_3$ from outside of Delhi is beyond the scope of this analysis and must also be taken into consideration by policy makers.

**Reviewer 1: Minor comments**

**Line 53**

**Here ozone is described as an important greenhouse gas in the mid-troposphere. However, ozone acts as a greenhouse gas throughout the depth of the troposphere, with a maximum radiative impact in the upper troposphere. See Figure 1 in the Supplement of Skeie et al., 2020.**

*Response:* Text has been changed to address that ozone is an important greenhouse gas throughout the troposphere, as seen in Skeie et al., 2020.

*Text change*: Tropospheric $O_3$ is both an air pollutant and an important greenhouse gas throughout the troposphere (Skeie et al., 2020).

**Line 64**

**"cocktail" is a fine analogy for conversational discussions, but not for a scientific paper. Use something like "range" instead.**

*Response*: "Cocktail" changed to "range" in text.

*Text change*: Unlike other pollutants such as $NO_x$ and $SO_2$, ground-level $O_3$ is not directly emitted but is formed in the atmosphere from the photochemical processing of a range of reactive precursor species (Calvert et al., 2015).

**Line 75**

**The presentation of basic ozone photochemistry should include a reference**

*Response*: Reference added.

*Text change*: In general, $O_3$ formation is mediated by the reactions of peroxy radicals, $RO_2$ and $HO_2$, formed in the OH-initiated oxidation of VOCs (R1), with NO to produce $NO_2$ (R2 and R4). $NO_2$ is then rapidly photolyzed back to NO, forming $O(^3P)$ (R5), which can rapidly react with $O_2$, leading to $O_3$ (R6). This recycling of NO to $NO_2$ leads to a net production of $O_3$ (Calvert et al., 2015).

**Line 96**

**Shouldn't but-2-enes be 2-butenes? Here is the relevant passage from Ran et al., 2011: "The most reactive species responsible for ozone formation are mainly alkenes and aromatics such as 2-butenes (18 %), isoprene (15 %), trimethylbenzenes (11 %), xylenes (8.5 %) and toluene (4.5 %)."**

*Response*: Although "2-butenes" is used in Ran et al., 2011, and elsewhere in the literature, the authors have kept the text as but-2-enes as this is the IUPAC preferred nomenclature. See

"Nomenclature of Organic Chemistry: IUPAC Recommendations and Preferred Names 2013, IUPAC Blue book, prepared for publication by Henri A Favre and Warren H Powell, by RSC Publishing, 2014 [ISBN 978-0-85404-182-4]; https://doi.org/10.1039/9781849733069".

**Line 104**

**A reference is needed for the statement on personal care products. McDonald et al. (2018) is a good one.**

*Response:* Reference added.

*Text change:* Oxygenated VOCs from solvent consumption and personal care products dominate the VOC-OH reactivity in recent years, leading to sustained high $O_3$ concentrations in the MCMA (McDonald et al., 2018, Zavala et al., 2020).

**Line 105**

**A reference is needed for this statement: "Understanding which precursor species are key to O3 production in any given city allows governments to introduce measures to combat air quality problems."**

*Response*: Reference added.

*Text change*: Understanding which precursor species are key to $O_3$ production in any given city allows governments to introduce measures to combat air quality problems (Molina, 2021).

**Line 135**

**"was attributed" should be "were attributed"**

*Response*: Changed.

*Text change*: However, increased concentrations of ground-level $O_3$ (> 10 %) were also observed and were attributed to reductions of NO leading to reduced consumption of $O_3$.

**Line 136**

**I don't think I've ever heard of the term "deweathered". Do you mean to say that meteorological biases were removed?**

*Response*: Changed.

*Text change*: Another study found that, after removing meteorological biases, concentrations of $NO_2$ and $PM_{2.5}$ at urban background sites in Delhi to have reduced by ~ 51 % and ~ 5 % respectively, with $O_3$ concentrations increasing by ~ 8 % (Shi et al., 2021).

**Line 182**

**high should be height**

*Response*: Changed.

*Text change:* IGDTUW facilitated the sampling of ambient air from a height of ~ 5 m and measurements were made of a large range of VOCs, o-VOCs, NOx, CO, $SO_2$, HONO, photolysis rates and PM.

**Line 367**

**Here you state that the observations "suggest" that the standard was exceeded on 16 days. But to say "suggest" implies that you aren't really sure. However, your measurements show that the standard was definitely exceeded on 16 days, and you should rephrase the sentence so that it reflects your confidence in your observations.**

*Response*: "suggest" has been changed to "show".

*Text change*: Our observations show that the 8-hour $O_3$ prescribed national standard is exceeded on 16 days during our 24-day measurement period (67 % of days)

**Line 369**

**Again, why use the word "suggest"? An official government document should clearly state the policy, with no ambiguity.**

*Response*: "suggest" has been changed to "state that"

*Text change*: The published national standards state that any pollutant which exceeds the prescribed values for two consecutive days qualifies for regular and continuous monitoring.

**Line 444**

**On line 348 the ozone peak is stated to occur at 13:00, but here the peak is stated to occur at 12:00. Please reconcile.**

*Response*: Thank you for spotting this – the peak occurs at 13:00. The text has been changed to reflect this.

*Text change*: In Delhi, $O_3$ concentrations rapidly increase from ~ 08:00, peaking around ~ 13:00.

**Line 495**

**There seems to be a typo in the following sentence in the caption to Figure 9: "The red diamond represents at point 1,1 represents modelled P(O3) at observed VOC and NOx concentrations."**

*Response*: Corrected.

*Text change*: The red diamond at point 1,1 represents modelled P($O_3$) at observed VOC and $NO_x$ concentrations.

**Line 573**

**Delete "the" before prevalence**

*Response*: "the" removed.

*Text change*: Urban NOₓ emissions (at tailpipe) are likely to decline over time, as a result of improved exhaust gas treatments, the turnover of the fleet to newer, less polluting vehicles, and the increasing prevalence of electric vehicles (Molina, 2020).

**Line 658**

**"represents and aggregate" should be "represents an aggregate"**

*Response*: Changed.

*Text change*: Whilst the OH reactivity analysis provides an instantaneous assessment of a VOC's reactivity, based on local observations with a small spatial footprint, the EDGAR Delhi-wide sectoral split represents an aggregate of a wider region and may under-represent the RT VOC contribution observed at the measurement site.

**References:**

Cooper, O. R., et al. (2020), Multi-decadal surface ozone trends at globally distributed remote locations, Elem Sci Anth, 8(1), p.23. DOI: http://doi.org/10.1525/elementa.420

Gaudel, A., et al. (2020), Aircraft observations since the 1990s reveal increases of tropospheric ozone at multiple locations across the Northern Hemisphere. Sci. Adv. 6, eaba8272, DOI: 10.1126/sciadv.aba8272

Liao, Z., Ling, Z., Gao, M., Sun, J., Zhao, W., Ma, P., Quan, J. and Fan, S., 2021. Tropospheric Ozone Variability Over Hong Kong Based on Recent 20 years (2000–2019) Ozonesonde Observation. Journal of Geophysical Research: Atmospheres, 126(3), p.e2020JD033054.

Lu, X., Zhang, L., Wang, X., Gao, M., Li, K., Zhang, Y., Yue, X. and Zhang, Y., 2020. Rapid increases in warm-season surface ozone and resulting health impact in China since 2013. Environmental Science & Technology Letters, 7(4), pp.240-247.

McDonald, B.C., De Gouw, J.A., Gilman, J.B., Jathar, S.H., Akherati, A., Cappa, C.D., Jimenez, J.L., Lee-Taylor, J., Hayes, P.L., McKeen, S.A. and Cui, Y.Y., 2018. Volatile chemical products emerging as largest petrochemical source of urban organic emissions. Science, 359(6377), pp.760-764.

Skeie, R.B., Myhre, G., Hodnebrog, Ø., Cameron-Smith, P.J., Deushi, M., Hegglin, M.I., Horowitz, L.W., Kramer, R.J., Michou, M., Mills, M.J. and Olivié, D.J., 2020. Historical total ozone radiative forcing derived from CMIP6 simulations. npj Climate and Atmospheric Science, 3(1), pp.1-10

Tarasick, D. W., et al. (2019), Tropospheric Ozone Assessment Report: Tropospheric ozone from 1877 to 2016, observed levels, trends and uncertainties. Elem Sci Anth, 7(1), DOI: http://doi.org/10.1525/elementa.376

---

## Author Comment (AC2)

The authors would like to thank the reviewers for their positive and constructive comments on this manuscript. Reviewer comments are in **bold**, followed by the author response and resulting text amendments.

**Reviewer 2: Comments**

**As the observed NOx and VOCs is very high in Delhi (much higher than previously published conditions in urban areas since 1990s), it is very interesting why the observed O3 concentrations were actually not high in Delhi. I compared the conditions of Dehli with that of Los Angels, and it is found that the current condition of Delhi is similar to that of 1970s in Los Angels (cf. Pollack et al., JGR, 2013). Nevertheless, in 1970s, the ozone concentrations were about 400 ppbv in Los Angels. So, it is actually quite useful for the authors to compare the observed conditions of air masses and the diagnosed ozone production rates and their controlling factors to other urban areas of the world (e.g. US, EU and China).**

*Response:* This is a very interesting point. Additional text has been added with respect to discussion on why ozone concentrations in Delhi are lower than Los Angeles in the 1970s, and more like observations in Beijing, Shanghai and Guangzhou, despite comparable VOC and $NO_x$ concentrations to Los Angeles.

*Text change:* Average mixing ratios of $NO_x$ and VOCs observed during the post-monsoon Delhi campaign are comparable to those observed in 1970s Los Angeles. High concentrations of $NO_x$ ($\approx$ 200 ppbV) and VOCs (e.g. benzene $\approx$ 10 ppbV; toluene $\approx$ 30 ppbV) led to large concentrations of $O_3$ in the US city, averaging at around 400 ppbV (Pollack et al., 2013). Despite the high $O_3$ concentrations observed in Delhi, $O_3$ concentrations in post-monsoon Delhi did not exceed 150 ppbV and are of similar magnitude to observed $O_3$ in Beijing, Shanghai and Guangzhou despite much higher observed VOC and $NO_x$ concentrations (Tan et al., 2019). The lower $O_3$ observed in Delhi compared to 1970s Los Angeles can be attributed to differences in both topography and meteorology. The isolated coastal city of Los Angeles is surrounded by mountains to the North and East, with the prevailing wind dominating from the coast. Owing to its basin-like topography, and with a cool on-shore sea breeze often creating a temperature inversion, the air mass circulates within the city and the transport of emissions out of the basin is impeded. Although landlocked Delhi lies to the southwest of the Himalayas, the city is very flat and resides far enough away from the mountain range to allow for the efficient transportation of air masses from the city. It is also important to consider that the very high concentrations of $NO_x$ and VOCs observed peak to comparable concentrations to Los Angeles during the evening and at night, where they are trapped owing to a shallow, stagnant boundary layer, and there is little to no photochemical activity. $O_3$ production rates in Delhi peaks in the morning, when concentrations of pollutants, though still high, are much lower than at night due to rapid boundary layer expansion (see section 3.4).

**Followed by comment 1, it would be useful to present the diurnal vacations of the diagnosed P(O3) in Section 3.4.**

*Response:* The diel profile of ozone production has been added to the supplementary and referenced in the text.

*Text change:* The modelled VOC-NO$_x$ P(O$_3$) isopleth supports the assignment of Delhi being, on average, in a VOC-sensitive photochemical regime (Sillman et al., 1990), with the diel profile of O$_3$ production peaking at 09:00 (see supplementary, figure S3).

**Section 3.5 and 3.6, the study of the VOCs sensitivities by class may be represented by a well established metrics in the study of ozone photochemistry - relative incremental reactivity (C.A. Cardelino and W.L. Chameides, J. Air & Waste Manage. Assoc. 1995).**

*Response:* Relative incremental reactivity has previously been considered by the authors as a metric to assess the O$_3$ production potential of VOCs. However, it was decided that changes in in situ O$_3$ production rate on incremental changes of individual VOCs was a better metric to use in this work, rather than comparing changes in O$_3$ concentrations as they provide a more meaningful quantification of the impact of changing O$_3$ production. The discussion of O$_3$ production, rather an O$_3$ concentration, is kept consistent throughout the text.

**Section 3.7, the study on the impact from the aerosol uptake and radiative forcing is very useful. Nevertheless, the study needs to be projected with more reality. With respect to the aerosol uptake effect, it can change the HO2 uptake rates when NO is small which is not the case for Delhi; but it can also change the HONO production rate which might be more important for Dehli in this case (high NO2 and high ASA). With respect to the change of the photolysis rates, the impact of aerosol could be complicated, the photolysis rates may be reduced in the near surface but enhanced in the higher place in the boundary layer when the aerosol SSA is high. The box model to diagnose the photochemistry processes is normally running with an assumption that the air mass is well mixed for the planetary boundary layer. Thus, the photolysis rates used in the model may be slightly different from the surface observations especially for the high aerosol atmosphere (i.e. Castro et al, AE, 2001). In short, the change of photolysis rates has to be much smaller than the current range and the discussion with that of aerosol shall be improved in this direction.**

*Response:* The text has been changed in line with the reviewers suggestions for clarifying the real impact of aerosol on photolysis rates and ultimately ozone production.

*Text change:* However, the impact of reducing aerosol on photolysis rates is complex. Reductions in aerosol at ground level may lead to either increased or decreased photolysis rates near the surface, depending on the scattering properties of the aerosol. Changes in photolysis rates from increased or

decreased aerosol loading may also vary throughout the depth of the boundary layer. The box model assumes photolysis rates are uniform throughout the boundary layer and that aerosols are well mixed. A more detailed study into the temporal and spatial patterns of aerosol in Delhi and its impact on photolysis rates is required to fully assess the aerosol impact on in situ $O_3$ production (Castro et al., 2001).

It is important to note that this study focuses on the sensitivity of $P(O_3)$ to ASA through $HO_2$ uptake only. Additional chemical consequences and feedbacks of decreasing aerosol, such as changes to HONO concentrations, have not been accounted for here. With the high levels of $NO_2$ observed in Delhi, the heterogenous conversion of $NO_2$ to HONO on particle surfaces may be an important mechanism (Liu et al., 2014; Lee et al., 2016; Tong et al., 2016; Lu et al., 2018b). HONO reductions from decreased ASA may lead to reduced OH radical formation in Delhi, thus the impact of ASA reduction on $P(O_3)$ may be underestimated in this work.

**The emission of monoterpene from anthropogenic sources is a new point worth to be highlighted. Even a paper Cash et al., 2021, in preparation is mentioned in the text some more description will be valuable also for this paper. The emissions from the process emission sector needs some more explains (e.g. which processes?).**

*Response:* A more detailed description of the potential sources of monoterpenes in Delhi has been added to the discussion of anthropogenic monoterpenes. The full breakdown of what constitutes as "process emissions" in the EDGAR inventory is detailed in the supplementary, and a reference to this has been added to the text. A more detailed analysis of the monoterpene sources in Delhi is beyond the scope of this work, and will be included in Cash et al., in preparation.

*Text change:* However, it is worth noting that the effectiveness of reducing the RT + PE source on modelled $P(O_3)$ is dependent on the proportion of anthropogenic monoterpene emissions in Delhi. According to the EDGAR emission inventory, 98.1% of anthropogenic monoterpenes in Delhi are attributed to process emissions (PE) (Table 2). These emissions include sources such as emissions from chemical industry, and other industrial processes, and include solvent emissions and emissions from product use (see Table S3 in supplementary). As emissions from these sources are grouped together in EDGAR inventory, the exact sources from which monoterpenes are attributed to cannot be identified. The sensitivity of $\Delta P(O_3)$ from reducing process emission sources (PE, RT+PE, and RT+PE+CM) is shown by the shaded regions in Figure 12, where the dashed lines represent the sensitivity limits where the observed monoterpenes are between 0% and 100% anthropogenic (as opposed to biogenic). There is relatively little impact on $P(O_3)$ on reducing RT+PE when monoterpenes are assumed to have an entirely biogenic source. However, it is clear that although the degree to which reducing process emissions along with road transport in this study impacts $P(O_3)$ cannot be accurately determined, even if no monoterpenes are reduced within this source, reducing it does not negatively impact $P(O_3)$. It is also important to consider possible under-estimations for the monoterpene contribution to RT. Monoterpene observations in Delhi were strongly correlated with CO emissions, suggesting anthropogenic sources (Stewart et al., 2021a). The EDGAR emissions inventory assigns 0% of the anthropogenic monoterpenes in the inventory to the RT source sector (table 2). An analysis of the PTR-QiTOF flux data, obtained at the IGDTUW measurement site directly after the concentration measurement period ended, suggests ~ 60% of the monoterpenes observed

could be attributed to traffic factors (Cash et al., 2021). A study by Wang et al, 2020 suggested vehicular and burning sources may contribute to the anthropogenic emissions of biogenic molecules. Other possible sources of anthropogenic monoterpenes in Delhi are emissions from cooking herbs and spices, and from fragrances and personal care products (Klein et al., 2016, McDonald et al., 2018).

**Reviewer 2: Technical comments**

**Indian megacity of Delhi, may be better writes like "the megacity Delhi, India", I think Delhi is a megacity also world wide.**

*Response:* Title has been changed.

*Text change:* In situ ozone production is highly sensitive to volatile organic compounds in Delhi, India.

**Cash et al., 2021, in preparation. should not be included in the reference list, it may be simply wrote in the line of 393 as (Cash et al., 2021, in preparation)**

*Response:* Reference removed from reference list.

**Figure 9, the isopleth can be improved if more sensitivity model runs are included**

*Response:* The authors are unsure on the level of detail the reviewer is requesting to improve the isopleth but have revised the text to provide the reader with more information using the isopleth data. A more detailed description on the isopleth has been added to the text. Suggestions for ways in which $NO_x$ and VOCs can be reduced to achieve the prescribed national ozone limits have been provided, making the analysis more useful for those interested in air quality management (see response to Reviewer 1). Using the isopleth data provided, the required reductions in VOCs to achieve ozone limits when $NO_x$ is reduced to 25%, 50% and 75% of observed values are detailed in the text.

*Text change:* As has been previously discussed, $O_3$ concentrations limits of 50 ppbV were regularly exceeding during the campaign, with the maximum daily 8-hour averages peaking at 88 ppbV (Figure 4). To successfully reduce $O_3$ to the limit of 50 ppbV, $O_3$ production must be reduced by 56%. This can be achieved by reducing $NO_x$ by 25%, 50% and 75% along with a concurrent reduction in VOCs of 48%, 61% and 78% respectively. To obtain a reduction in $O_3$ production without reducing VOCs would require a $NO_x$ reduction of at least 92%.